# Inhibition of *O*-GlcNAc transferase activates type I interferon-dependent antitumor immunity by bridging cGAS-STING pathway

Jianwen Chen[1,2†], Bao Zhao[1,2†], Hong Dong[1†], Tianliang Li[1], Xiang Cheng[1,2], Wang Gong[3], Jing Wang[2,4], Junran Zhang[2,5], Gang Xin[1,2], Yanbao Yu[6], Yu L Lei[3], Jennifer D Black[7], Zihai Li[2], Haitao Wen[1,2]*

[1]Department of Microbial Infection and Immunity, Infectious Disease Institute, The Ohio State University, Columbus, United States; [2]Pelotonia Institute for Immuno-Oncology, The Ohio State University Comprehensive Cancer Center, The Ohio State University, Columbus, United States; [3]Department of Periodontics and Oral Medicine, University of Michigan School of Dentistry, University of Michigan Rogel Cancer Center, University of Michigan, Ann Arbor, United States; [4]Department of Cancer Biology and Genetics, The Ohio State University, Columbus, United States; [5]Department of Radiation Oncology, The Ohio State University, Columbus, United States; [6]Department of Chemistry and Biochemistry, University of Delaware, Newark, United States; [7]Eppley Institute for Research in Cancer and Allied Diseases, University of Nebraska Medical Center, Omaha, United States

*For correspondence:
Haitao.Wen@osumc.edu

†These authors contributed equally to this work

Competing interest: The authors declare that no competing interests exist.

**Abstract** The *O*-GlcNAc transferase (OGT) is an essential enzyme that mediates protein *O*-GlcNAcylation, a unique form of posttranslational modification of many nuclear and cytosolic proteins. Recent studies observed increased OGT and *O*-GlcNAcylation levels in a broad range of human cancer tissues compared to adjacent normal tissues, indicating a universal effect of OGT in promoting tumorigenesis. Here, we show that OGT is essential for tumor growth in immunocompetent mice by repressing the cyclic GMP-AMP synthase (cGAS)-dependent DNA sensing pathway. We found that deletion of OGT ($Ogt^{-/-}$) caused a marked reduction in tumor growth in both syngeneic mice tumor models and a genetic mice colorectal cancer (CRC) model induced by mutation of the *Apc* gene ($Apc^{min}$). Pharmacological inhibition or genetic deletion of OGT induced a robust genomic instability (GIN), leading to cGAS-dependent production of the type I interferon (IFN-I) and IFN-stimulated genes (ISGs). As a result, deletion of *Cgas* or *Sting* from $Ogt^{-/-}$ cancer cells restored tumor growth, and this correlated with impaired CD8+ T-cell-mediated antitumor immunity. Mechanistically, we found that OGT-dependent cleavage of host cell factor C1 (HCF-1) is required for the avoidance of GIN and IFN-I production in tumors. In summary, our results identify OGT-mediated genomic stability and activate cGAS-STING pathway as an important tumor-cell-intrinsic mechanism to repress antitumor immunity.

## eLife assessment

The author demonstrates that deficiency or pharmacological inhibition of O-glcNac transferase (OGT) enhances tumor immunity in colorectal cancer models. This **useful** study unveils that OGT deficiency triggers a DNA damage response that can affect immune status in colorectal cancers. It provides **convincing** evidence showing that OGT-mediated processing of HSF1 is crucial in maintaining genomic integrity.

## Introduction

Cancer cells can maintain malignant phenotypes partially due to altering the post-translational modification (PTM) patterns of cancer-related functional proteins under the stimulation of extracellular and intracellular factors (*Liu et al., 2022*). Protein modification by the *O*-linked β-N-acetylglucosamine (*O*-GlcNAc) is a dynamic and reversible post-translational modification, which is added to the hydroxyl group of a specific serine or threonine residue in a target protein by *O*-GlcNAc transferase (OGT) and removed by *O*-GlcNAcase (OGA; *Li et al., 2018*). *O*-GlcNAcylation is a fast-cycle and nutrient-sensitive PTM, which modifies thousands of cytoplasmic, nuclear, and mitochondrial proteins and mediates crosstalk with protein phosphorylation, regulating signal transduction and affecting protein localization, activity, stability, and protein-protein interaction. Dysregulation of *O*-GlcNAcylation is associated with multiple metabolic diseases and cancer (*Li et al., 2019*; *Zhu and Hart, 2023*).

Recent studies observed increased OGT and *O*-GlcNAcylation level in human colon cancer tissues compared to adjacent normal tissues (*Mi et al., 2011*; *Phueaouan et al., 2013*; *Olivier-Van Stichelen et al., 2014*), indicating an essential role of OGT-mediated protein *O*-GlcNAcylation in the pathogenesis of colon cancer. Several oncogenic proteins that are involved in the pathogenesis of colon cancer have been shown to be directly modified by *O*-GlcNAc, including β-catenin and NF-κB (*Fardini et al., 2013*; *Chaiyawat et al., 2014*; *Ozcan et al., 2010*). For example, *O*-GlcNAcylation of β-catenin at T41 inhibits its phosphorylation, which subsequently attenuates its ubiquitination and degradation and promotes oncogenic activity (*Olivier-Van Stichelen et al., 2014*; *Olivier-Van Stichelen et al., 2012*). In contrast, inhibitory roles of *O*-GlcNAc signaling in the growth of human colon cancer and in oncogenic Wnt/β-catenin signaling have also been reported (*Yehezkel et al., 2012*; *Wu et al., 2014*). The aforementioned studies show that *O*-GlcNAc modification of specific proteins can play opposing roles in tumorigenesis. However, the overall effect of OGT-mediated *O*-GlcNAcylation in cancer remains unknown.

The cGAS/STING cytosolic DNA-sensing pathway plays a vital role in activating the innate immune response and production of the type I interferons (IFN-I) (*Li et al., 2023*). Cyclic guanosine monophosphate (GMP)–adenosine monophosphate (AMP) synthase (cGAS) interacts with cytosolic double-stranded DNA (dsDNA) in a sequence-independent manner. The direct binding of cGAS to cytosolic dsDNA promotes cGAS homodimerization and activates the catalytic activity of cGAS, producing 2′, 3′-cyclic GMP-AMP (cGAMP) from ATP and GTP. The second messenger cGAMP binds to and activates the endoplasmic resident stimulator of interferon genes (STING). Once activated, STING translocates to the ER-Golgi intermediate compartment (ERGIC) to recruit TANK-binding kinase 1 (TBK1) and IFN regulatory factor 3 (IRF3), leading to the production of IFN-I and activation of numerous IFN-stimulated genes (ISGs; *Ghosh et al., 2021*; *Fang et al., 2023*; *Kwon and Bakhoum, 2020*). However, it remains largely unknown whether OGT expression affects cGAS-STING pathway and antitumor immunity.

In this study, we find that deficiency or pharmacological inhibition of OGT and subsequent accumulation of cytosolic dsDNA activates the cGAS-STING pathway and induces CD8$^+$ T-cell-dependent antitumor immunity. Deletion of cGAS or STING diminishes DNA sensing and lead to progressive tumor growth. Mechanistically, we show that OGT could interacts with HCF-1 and cleaves it, which contributes to the maintenance of genomic stability. Re-expression of HCF-1$^{C600}$ in $Ogt^{-/-}$ tumor cells inhibit production of cytosolic dsDNA and IFN-I. In summary, our findings demonstrate that OGT-mediated DNA damage and activate cGAS-STING pathway as an important tumor cell-intrinsic mechanism to repress antitumor immunity and provides a window for potential therapeutic opportunities for in OGT-dependent cancer.

## Results

### Increased OGT expression in human and mouse tumor samples

Protein *O*-GlcNAcylation is upregulated in various cancers (*Yehezkel et al., 2012*; *Gu et al., 2010*; *Itkonen et al., 2013*; *Jiang et al., 2019*; *Yi et al., 2012*). OGT is the only known enzyme that mediates *O*-GlcNAcylation of proteins at the Ser or Thr residues (*Yang and Qian, 2017*), we hypothesized that OGT could serve as an important regulator to regulate cancer cell growth and serve as a biomarker for cancer. Initial analysis of data from The Cancer Genome Atlas (TCGA) dataset in GEPIA2 (http://gepia2.cancer-pku.cn/#index) and found a significant positive correlation between *OGT* mRNA expression

and tumorigenesis in bladder urothelial carcinoma (BLCA), cholangiocarcinoma (CHOL), colon adeno-carcinoma (COAD), esophageal carcinoma (ESCA), head and neck squamous cell carcinoma (HNSC), kidney chromophobe (LIHC), lung adenocarcinoma (LUAD), prostate adenocarcinoma (PRAD), rectum adenocarcinoma (READ), sarcoma (SARC), and a stomach adenocarcinoma (STAD) (*Figure 1—figure supplement 1A*). Next, we utilized UALCAN database analysis, and found that *OGT* mRNA expression was compared between 286 COAD samples and 41 adjacent or normal samples, the expression of *OGT* was significantly increased in COAD at the transcriptional level (*Figure 1A*). A significant positive correlation tendency between *OGT* mRNA expression and individual different tumor stages was observed. The stage IV COAD tissues exhibited the highest expression level of *OGT* in compared with low stage (*Figure 1B*). We also observed a significant positive correlation between *OGT* mRNA expression and nodal metastasis status (*Figure 1C*). Furthermore, we found that the protein level of OGT was also significantly increased in COAD patient samples and amongst individual different stages based on the CPTAC and HPA online database (*Figure 1D–F*). Similar results are found in LUAD (*Figure 1—figure supplement 1B-F*). These results suggest that high mRNA and protein levels of OGT in tumorigenesis were consistent in different databases.

To determine whether intestinal OGT expression was increased in *Apc^min* colorectal tumor mouse model, intestinal tissues were collected for western blot and immunohistochemical staining (IHC) analysis. Intestinal OGT protein was markedly increased in *Apc^min* mouse tumor tissues compared to adjacent or normal tissues (*Figure 1G*). As expected, *O*-GlcNAcylation proteins levels in intestinal tissues were also significantly higher in tumor tissues than adjacent or normal tissues. IHC staining revealed that mouse OGT protein was markedly higher in tumor tissues than in adjacent tissues (*Figure 1H*), which was consistent with the expression pattern in human samples. OGT protein was also upregulated in the azoxymethane (AOM)/dextran sodium sulfate (DSS) colorectal tumor model (*Figure 1I–K*). Together, these data strongly suggest that OGT may play a critical role in tumorigenesis and serve as a prognostic marker and therapeutic target in cancer treatment.

## Epithelial OGT deletion inhibits mouse colorectal tumorigenesis

To determine whether OGT may be an important therapeutic target for tumor treatment, we generated an intestinal epithelial cell-specific *Ogt* deletion (*Ogt* IEC cKO) by crossing *Ogt^fl/fl* mice with Villin-Cre mice. *Ogt* IEC cKO mice were then crossed with *Apc^min* mice to generate *Ogt* IEC cKO *Apc^min* mice (*Figure 2—figure supplement 1*). To determine whether intestinal OGT expression was decreased in *Ogt* IEC cKO *Apc^min* mice, intestinal tissues were collected for western blot analysis. Intestinal OGT levels were drastically decreased in *Ogt* IEC cKO *Apc^min mice compared to wildtype Apc^min mice* (*Figure 2A*). Importantly, deletion of the *Ogt* in intestinal epithelial cells resulted in significantly reduced tumor size and total number of polyps at 20 weeks of age (*Figure 2A*), indicating that OGT promotes oncogenic transformation in colorectal tumor in vivo. To gain insight into the role of protein on intestinal carcinogenesis in intestinal cancer, we examined the differences in intestinal carcinogenesis. Hematoxylin and eosin (H&E) staining demonstrated that epithelial inflammation increased in *Ogt* IEC cKO *Apc^min* mice compared to wildtype *Apc^min* mice (*Figure 2B*). OGT deficiency in intestinal tissues was also associated with significantly elevated gene expression of key pro-inflammatory cytokines like interleukin *Il1a*, *Il6*, *TNFa*, as well as several interferons and ISGs like *Isg15*, *Mx1*, *Cxcl10* (*Figure 2C*). Furthermore, we found IL-2, IL-6, IL-10, IL-17, IFN-α, IFN-β, IFN-γ, and CXCL10 were upregulated compared with control mice (*Figure 2D*).

## OGT deficiency activates cGAS/STING-dependent IFN-I pathway

Based on previous studies, we found that several interferons and several interferon-stimulated genes, such as *Isg15*, *Mx1*, *Ifna4*, *Ifnb1*, and *Cxcl10* were upregulated. We hypothesized that OGT deficiency may activate the type I IFN pathway. First, we generated *Ogt* knock outs of murine colorectal carcinoma models like MC38, LLC, B16-OVA and of a human colorectal cell line, HT29. We then evaluated the expression changes of *Ifna4*, *Ifnb1*, *Isg15*, *Mx1* and found that all the genes except *Ifna4* mRNA expression were significantly increased in all cultured *Ogt* knockout cells (*Figure 3A–D*). Next, we investigated how the type I IFN pathway in *Ogt* knockout cells was activated (*McNab et al., 2015*; *Liu et al., 2015*). We found that while the phosphorylation of STAT1, TBK1 and IRF3 is increased, the STING expression is reduced in *Ogt* knockout MC38, LLC, B16-OVA, and HT29 cells (*Figure 3E–F*). To determine whether this effect is specific for *Ogt* knockout, we stably expressed exogenous OGT

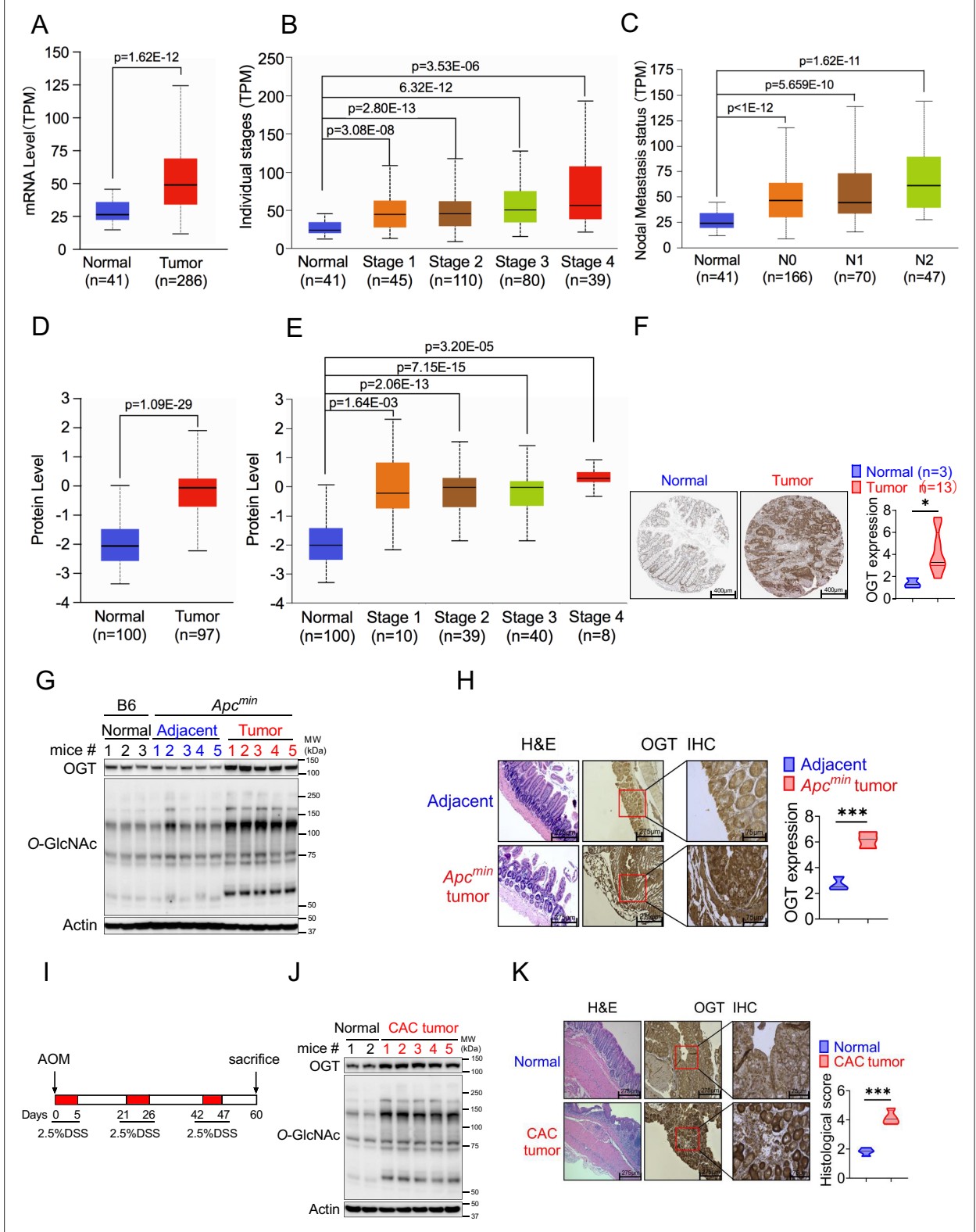

**Figure 1.** OGT is significantly upregulated in human and mouse tumor samples. (**A–C**) Boxplot showing mRNA expression level of *Ogt*, Normal and tumor samples (**A**), Individual stages (**B**), Nodal metastasis status (**C**). The plot was generated using the UALCAN online server. (**D–E**) Boxplot showing protein expression level of OGT, Normal and tumor samples (**A**), Individual stages (**B**). The plot was generated using the UALCAN online server (https://ualcan.path.uab.edu/analysis.html). (**F**) IHC analysis of OGT expression in normal colon tissues, primary colon tumor samples (from Human Protein Atlas, https://www.proteinatlas.org/), scale bar: 400 µm. (**G**) Western blot analysis of OGT and *O*-GlcNAc expression in normal, adjacent and tumor tissues in

*Figure 1 continued on next page*

*Figure 1 continued*

*Apc^min* spontaneous tumor mice. (**H**) HE and IHC staining of OGT in adjacent and tumor tissues in *Apc^min* spontaneous tumor mice, scale bar: left panel 275 µm, right panel 75 µm, n=3 respectively. (**I**) Schematic of AOM/DSS model of colitis-associated colorectal cancer (CAC). (**J**) Western blot analysis of OGT and *O*-GlcNAc expression in normal, adjacent and tumor tissues in CAC model. (**K**) HE and IHC staining of OGT in adjacent and tumor tissues in CAC model, scale bar: left panel 275 µm, right panel 75 µm, n=4 respectively. human samples (**A–F**), mouse samples (**G–K**). Statistical significance was determined by Pearson test, unpaired Student's t-test, *p<0.05, **p<0.01, ***p<0.001, ns, no significant difference. Data represent the mean of ± SD.

The online version of this article includes the following figure supplement(s) for figure 1:

**Figure supplement 1.** The expression pattern of OGT in TCGA and CTPAC databases.

in *Ogt* knockout cells and found that the type I IFN pathway activation effect is abolished (**Figure 3F**). The type I IFN pathway is typically activated by both viral RNA and dsDNA. In order to eliminate the type I IFN pathway activate independent on RNA but dependent on dsDNA, we knocked out MAVS protein in *Ogt^{-/-}* cells, which is a RNA pivotal adaptor protein activate the downstream protein kinase TBK1, IFN pathway and ISGs in *Ogt^{-/-}* cells (*24*), and we found that knockout of MAVS in *Ogt^{-/-}* cells has no effect on either *Isg15, Mx1, Ifna4* and *Ifnb1* expression or the type I IFN pathway (**Figure 3G–H**).

We further investigated the mechanism of the type I IFN signaling activation and found that *Ogt* knockout induced the reduction of the STING. This result is similar to previous studies (*Gui et al., 2019*; *Gonugunta et al., 2017*). The phenotype of *Ogt* knockout implies that STING is involved in the activation of the type I IFN pathway. We hypothesized that OGT negatively regulates type I

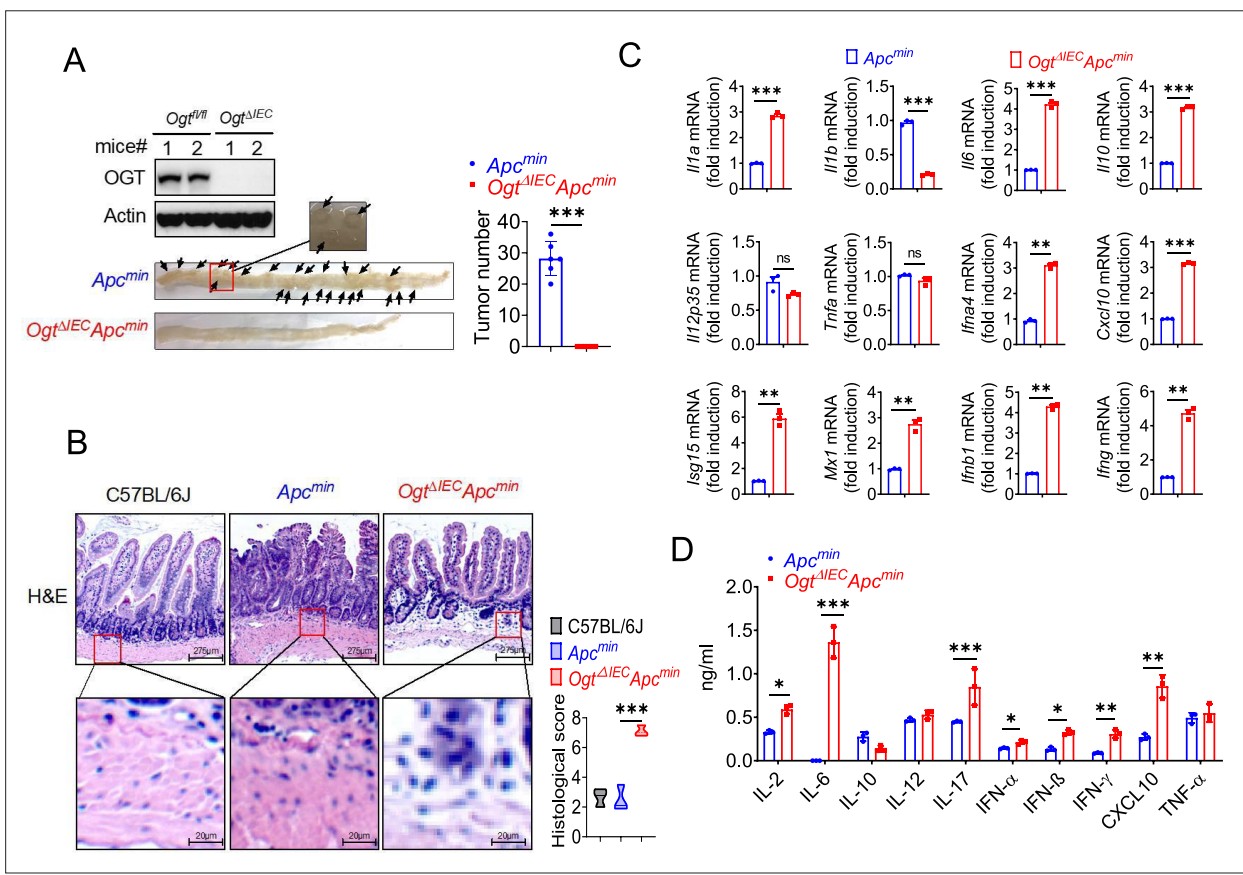

**Figure 2.** Epithelial OGT deletion inhibits mouse colorectal tumorigenesis. (**A**) Western blot analysis of OGT expression in intestinal tissues and counting tumor numbers in APC^min and *Ogt* IEC cKO mice. (**B**) Histology analysis of intestinal carcinogenesis by HE staining, scale bar: up panel 275 µm, bottom panel 20 µm, n=3, respectively. (**C**) Real-time PCR analysis of cytokines mRNA expression in intestine. (**D**) ELISA analysis of cytokines expression in intestine. Statistical significance was determined by unpaired Student's t-test, *p<0.05, **p<0.01, ***p<0.001, ns, no significant difference. Data represent the mean of ± SD.

The online version of this article includes the following figure supplement(s) for figure 2:

**Figure supplement 1.** The genotype of *APC^min*, *Villin-Cre* and *Ogt^{fl/fl}* mice.

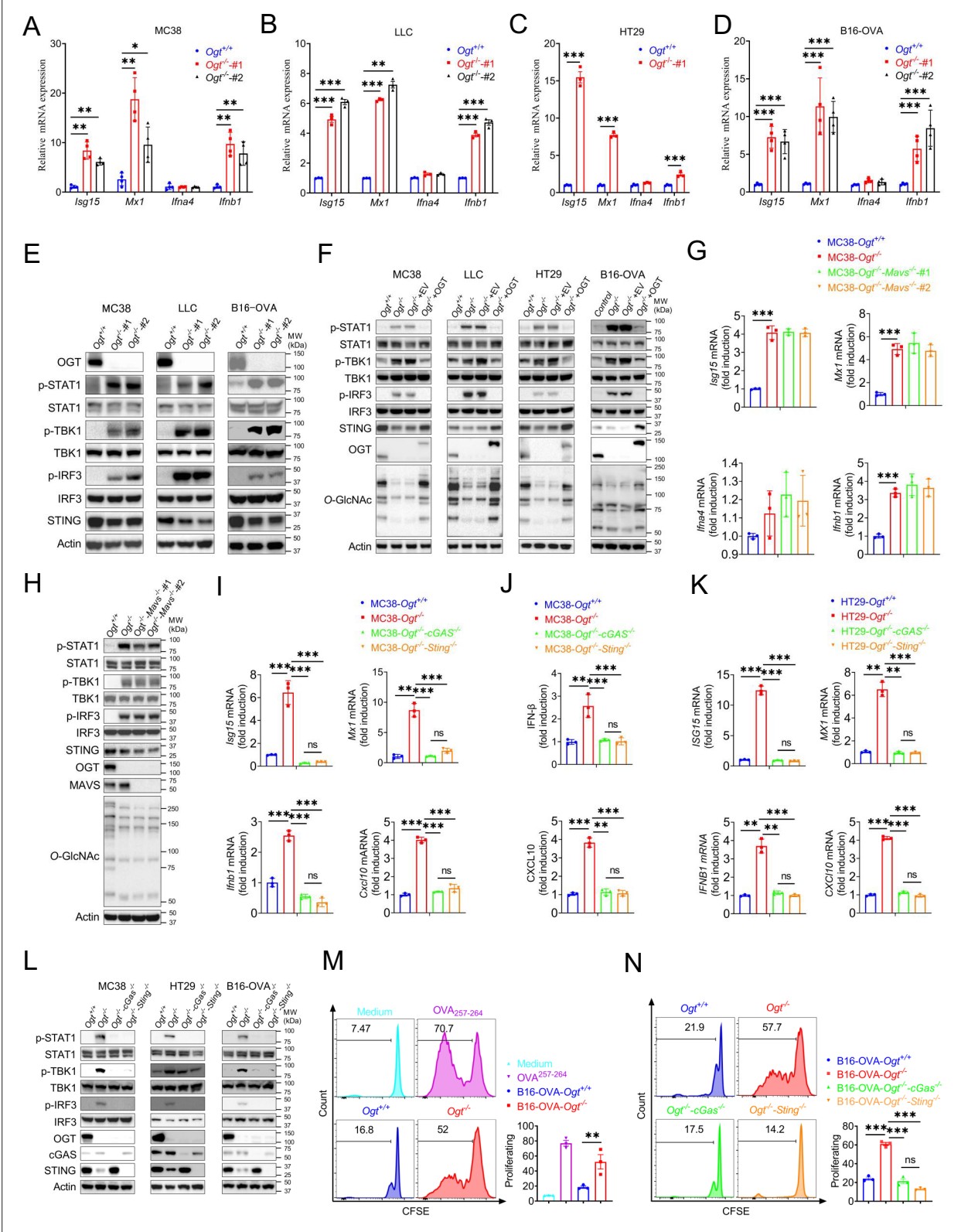

**Figure 3.** OGT deficiency induces cGAS/STING-dependent the type I IFN pathway. (**A–D**) Real-time PCR analysis of cytokines mRNA expression in different *Ogt⁻/⁻* cell lines including MC38 (**A**), LLC (**B**), HT29 (**C**), B16-OVA (**D**) cells. (**E**) Western blot analysis of the activation of the interferon signaling pathway in different *Ogt⁻/⁻* cell lines including MC38, LLC and B16-OVA cells. (**F**) Western blot analysis of the activation of the interferon signaling pathway in *Ogt⁻/⁻* rescued cell lines including MC38, LLC, HT29 and B16-OVA cells. (**G–H**) Real-time PCR and western blot analysis of cytokines mRNA

*Figure 3 continued on next page*

*Figure 3 continued*

expression and the activation of the interferon signaling pathway in different $Ogt^{-/-}Mavs^{-/-}$ double knockout clones in MC38 cells. (**I–K**) Real-time PCR and ELISA analysis of cytokines mRNA expression in $Ogt^{-/-}cGAS^{-/-}$ double knockout clones in MC38 (**I–J**), HT29 (**K**) cells. (**L**) Western blot analysis of the activation of the interferon signaling pathway in $Ogt^{-/-}cGAS^{-/-}$ or $Ogt^{-/-}Sting^{-/-}$ double knockout clones in MC38, HT29 and B16-OVA cells. (**M–N**) BMDCs pre-treated with B16-OVA-$Ogt^{-/-}$ (**L**), B16-OVA-$Ogt^{-/-}cGAS^{-/-}$ or B16-OVA-$Ogt^{-/-}Sting^{-/-}$ cells (**M**) supernatant, and co-cultured with OT-1 T cell, then T cell proliferation was evaluated by flow cytometry, OVA$^{257-264}$ as a positive control. Representative fluorescence-activated cell sorting histograms and statistical data are shown. Data are representative of two or three independent experiments. Statistical significance was determined by unpaired Student's t-test, one-way ANOVA, two-way ANOVA, *p<0.05, **p<0.01, ***p<0.001, ns, no significant difference. Data represent the mean of ± SD.

The online version of this article includes the following figure supplement(s) for figure 3:

**Figure supplement 1.** In vitro Cross-Priming of T Cells by $Ifnar^{-/-}$ BMDCs.

IFN through the cGAS/STING pathway. Next, we produced the OGT/cGAS and OGT/STING double knockout cells both in MC38, HT29 and B16-OVA. Surprisingly, these type I IFN and *Cxcl10* signals disappeared in OGT/cGAS and OGT/STING double knockout cells both in mRNA and protein level (*Figure 3I–L*).

The type I IFN signaling is a key pathway that promotes antigen presentation and DC activation (*Zhu et al., 2019*; *Joffre et al., 2012*; *Zitvogel et al., 2015*). Phagocytosis of extracellular tumor DNA by DCs triggers the activation of the cGAS-STING-IFN pathway (*Woo et al., 2014*; *Xu et al., 2017*). To determine whether the increased cGAS-STING-IFN signaling in *Ogt* knockout cells provides an activated signal for single epitope-specific CD8$^+$ T cell priming by antigen-presenting cells (APCs), we added the supernatant from B16-OVA tumor cells into the co-culture system of BMDCs and OT-I cells. The results showed that BMDCs pre-cultured with $Ogt^{+/+}$ or $Ogt^{-/-}$ B16-OVA tumor cells supernatant provided a potent activated signal for optimal single epitope-specific T cell proliferation (*Figure 3M*), while cGAS or STING deficiency in $Ogt^{-/-}$ tumor cells diminished such effects (*Figure 3N*). Further, we added the supernatant from $Ogt^{+/+}$ or $Ogt^{-/-}$ B16-OVA tumor cells into the co-culture system of *Ifnar1* knockout BMDCs and OT-I cells. The cell proliferation was abolished when treated with $Ifnar1^{-/-}$ BMDCs pre-cultured with $Ogt^{+/+}$ or $Ogt^{-/-}$ B16-OVA tumor cells supernatant. (*Figure 3—figure supplement 1*). These results demonstrated that *Ogt* deficiency activate antitumor CD8$^+$ T cells response is dependent on the type I IFN signal in dendritic cells and in a manner dependent on the tumor-cell-intrinsic cGAS-STING pathway.

## OGT deficiency causes DNA damage and cytosolic DNA accumulation

Because the accumulation of cytosolic DNA is a consequence of nuclear DNA damage, which can activate immune response. The cGAS senses cytoplasmic DNA as a consequence of nuclear DNA damage (*Li and Chen, 2018*). Previous studies showed that the STING is degraded while cGAS-STING is activated (*Gui et al., 2019*). As mentioned above, the STING degraded in *Ogt* knockout cells. We hypothesized that whether the cytosolic DNA is accumulated in *Ogt* knockout cells, we stained cytosolic double-strand DNA (dsDNA) with PicoGreen, a widely used immunofluorescence staining that selectively binds to dsDNA (*Shen et al., 2015*; *Lu et al., 2021*), and found that a significantly higher percentage of *Ogt* deficiency cells than control cells in two different *Ogt* knockout MC38 and LLC cell clones, respectively (*Figure 4A* and *Figure 4—figure supplement 1A*). In addition, we also stained dsDNA with anti-dsDNA immunofluorescence and the results were similar to PicoGreen staining (*Figure 4B* and *Figure 4—figure supplement 1B*). We next assessed the phosphorylation of H2AX at Ser 139 (γH2AX), an indirect marker of DNA DSBs in the cell lines (*Chowdhury et al., 2005*). Comparable expression levels of H2AX, we found that γH2AX was dramatically increased in *Ogt* knockout MC38, LLC, HT29 and B16-OVA cells (*Figure 4C and F*). We also used the anti-γH2AX immunofluorescence staining and comet assays, a classical quantifying and analyzing DNA damage, and found that *Ogt* knockout significantly induced immense DNA strand breakage in MC38 cells (*Figure 4D–E*). Finally, the rescued results showed that exogenous OGT expression abolished the γH2AX expression both in MC38, LLC, HT29 and B16-OVA cells and DNA damage in MC38 cells (*Figure 4F–H*). Together, these data indicated that *Ogt* knockout caused DNA damage and induced the cytosolic DNA accumulation.

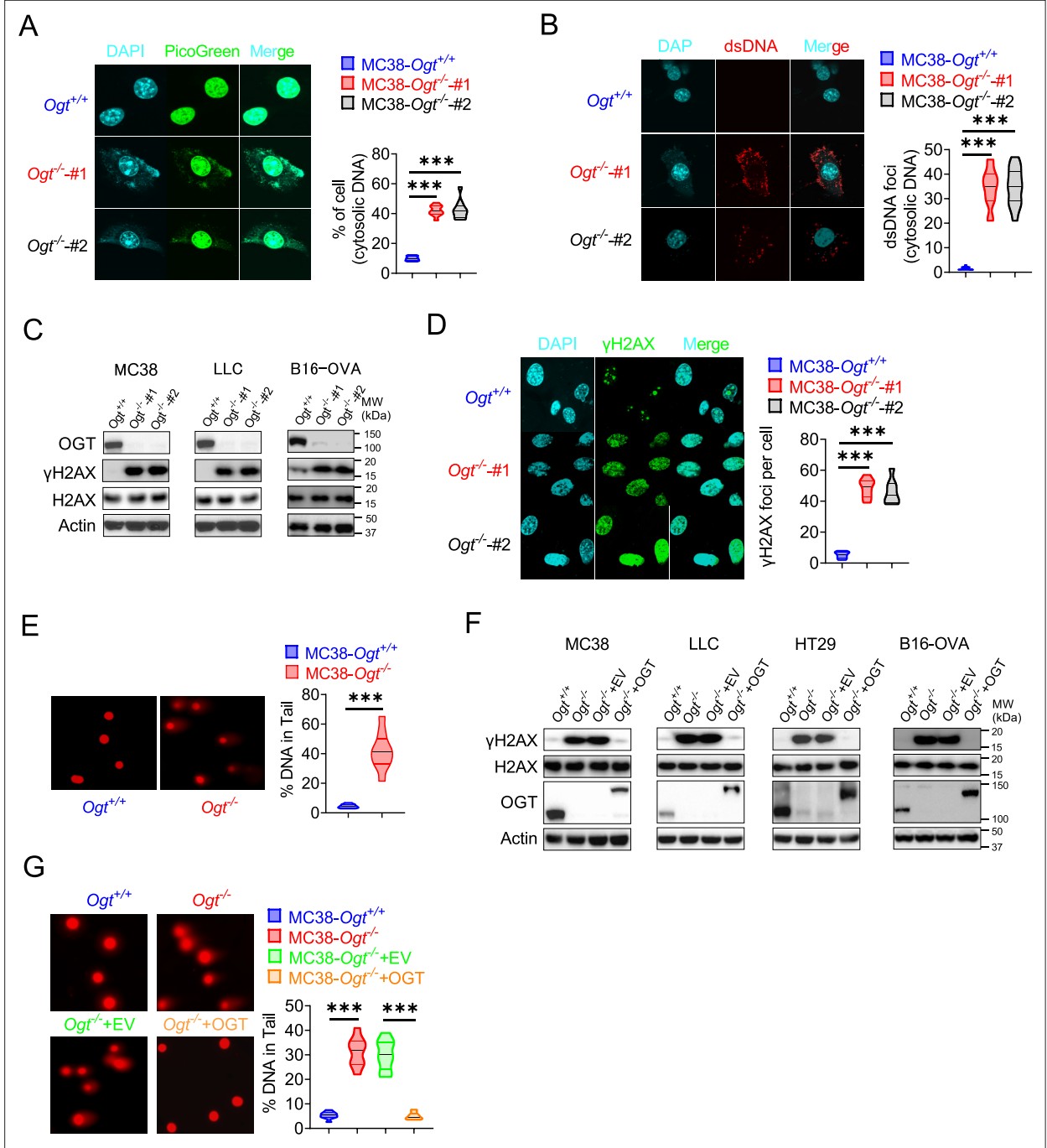

**Figure 4.** OGT deficiency causes DNA damage and accumulates cytosolic DNA. (**A**) The extranuclear dsDNA in different $Ogt^{-/-}$ MC38 cells clones were determined by PicoGreen staining assay was quantified by image J. (**B**) The extranuclear dsDNA in different $Ogt^{-/-}$ MC38 cells clones were determined by anti-dsDNA fluorescence staining assay and was quantified by image J. (**C**) Western blot analysis of γH2AX and H2AX expression in different $Ogt^{-/-}$ cell lines including MC38, LLC and B16-OVA cells. (**D**) Analysis of γH2AX and H2AX expression in different $Ogt^{-/-}$ clones by anti-γH2AX staining assay and was quantified by image (J). (**E**) The DNA damage was determined by comet assay, and extranuclear dsDNA was analyzed by using CometScore in $Ogt^{-/-}$ MC38 cells. (**F**) Western blot analysis of γH2AX and H2AX expression in $Ogt^{-/-}$ rescued cells including MC38, LLC, HT29 and B16-OVA cells. (**G**) The DNA damage in rescued MC38 cells were determined by comet assay, and extranuclear dsDNA was analyzed by using CometScore. Data are representative of three or four independent experiments. Statistical significance was determined by unpaired Student's t-test, one-way ANOVA, *$p<0.05$, **$p<0.01$, ***$p<0.001$, ns, no significant difference. Data represent the mean of ± SD.

The online version of this article includes the following figure supplement(s) for figure 4:

**Figure supplement 1.** OGT deficiency causes DNA damage and accumulates cytosolic DNA.

## The C-terminus of HCF-1 rescues DNA damage and IFN-I pathway in $Ogt^{-/-}$ cells

Previous studies in *Figure 3F* and *Figure 4F* showed that expression of exogenous OGT abolished the type I IFN and γH2AX signals. To explore the mechanism how OGT regulate the IFN and γH2AX production. Next, we use GFP agarose immunoprecipitation and liquid chromatography coupled to tandem MS (LC-MS/MS) to identify different proteins that interacted with OGT in human colonic cancer cell line HT29. Interestingly, HCF-1 was the most enriched protein in the precipitates from *OGT* restored HT29 knockout cells compared to *OGT* knockout cells reconstituted with empty vector, based on the number of peptides (indicating the identification confidence) and the number of peptide-spectrum matches (PSMs, indicating the abundance; *Figure 5A* and *Supplementary file 2*). Previous studies showed that the human epigenetic cell-cycle regulator (HCF-1) undergoes an unusual proteolytic maturation by OGT cleavage and process resulting in stably associated HCF-1$^{N1011}$ and HCF-1$^{C600}$ subunits that regulate different aspects of the cell cycle (*Capotosti et al., 2011*). We found similar results in this study by using co-immunoprecipitation assay (*Figure 5B–C*). We rescued the HCF-1 cleavage phenotype by expression of exogenous OGT (*Figure 5D*). To clarify how to OGT regulate function of HCF-1, we transfected empty vector (EV), the full-length HCF-1-HA (HCF-1$^{FL}$), HCF-1$^{1-1011}$-HA (HCF-1$^{N1011}$), HCF-1$^{1-450}$-HA (HCF-1$^{N450}$), HCF-1$^{450-1011}$-HA (HCF-1$^{N450-1011}$) and HCF-1$^{1436-2035}$-HA (HCF-1$^{C600}$) and myc-OGT and found that OGT can bind HCF-1$^{C600}$ (*Figure 5E*), furthermore OGT directly bind the HCF-1$^{C600}$ (*Figure 5—figure supplement 1A, B*). To further understand the physiological function of HCF-1, we transfected EV, HCF-1$^{FL}$, HCF-1$^{N1011}$and HCF-1$^{C600}$ into MC38 control and OGT knockout cells, respectively. As shown in *Figure 5F*, transfected HCF-1$^{FL}$ and HCF-1$^{N1011}$ didn't rescue the gene expression differences, such as *Isg15, Mx1* and *Ifnb1*, but HCF-1$^{C600}$ can restore the gene expression differences. These results were confirmed by using western blot and anti-dsDNA immunofluorescence staining (*Figure 5G–H*). Overall, these data indicated that OGT regulates the HCF-1 cleavage and maturation, HCF-1$^{C600}$ can eliminate the cytosolic DNA accumulation, DNA damage, the type I IFN activation and restrain cGAS-STING-mediated DNA sensing.

## OGT deficiency inhibits tumor progression through enhancing infiltration by CD8$^+$ T cells

We examined whether inhibiting *Ogt* can delay the tumor growth and prolong the survival. To further verify these effects, we utilized transplanted tumor model. We found that the *Ogt* deficiency MC38 colorectal tumor cells have no obvious inhibit cell growth in vitro (*Figure 6—figure supplement 1A*). However, we found that a significant delay in tumor growth, tumor weight and prolong mice survival compared to the control group (*Figure 6A–B*). We also used lewis lung carcinoma (LLC) cells and the B16-OVA melanoma cells because they represent an aggressive murine tumor model and are highly resistant to various immunotherapies, similar results were shown in *Figure 6C–D* and *Figure 6—figure supplement 1B-E*. These results showed that *Ogt* deficiency could delay tumor growth and prolong survival of mice in MC38, LLC, and B16-OVA tumor model. We next examined whether the enhanced antitumor function is related to tumor microenvironment (TME). Consistently, tumors from C57BL/6 immunocompetent mice bearing $Ogt^{-/-}$ MC38 tumors compared with MC38 control tumors, showed higher proportion of CD8$^+$ and CD4$^+$ T cells, functional CD8$^+$ IFN-γ$^+$, CD8$^+$ TNF-α$^+$ and CD8$^+$ IFN-γ$^+$ TNF-α$^+$ double positive T cells, but not CD45$^+$, CD11b$^+$ CD11c$^+$, CD11b$^+$ F4/80$^+$, CD11b$^+$ Ly6C$^+$ and Treg cells (*Figure 6E–H* and *Figure 7—figure supplement 1D-H*). The similar results were also observed in mice challenged with LLC or B16-OVA cells (*Figure 7—figure supplement 1I-P*).

To determine whether *Ogt* deficiency is dependent upon adaptive immune system, we inoculated MC38 control and *Ogt* knockout cells into immunodeficient $Rag2^{-/-}$ mice and tracked tumor growth. The difference of tumor growth rate disappeared between MC38-*Ogt* knockout and control ones (*Figure 6I*), indicating their association with an impaired immune response. We then postulated that *Ogt* knockout might have potent antitumor effects in vivo through CD4$^+$ T and CD8$^+$ T cells. In order to test this hypothesis, we performed antibody-mediated CD4$^+$ T or CD8$^+$ T cells depletion in $Ogt^{-/-}$ and control tumor-bearing mice and examined the tumor growth and survival. Depletion CD8$^+$ T cells dramatically enhanced the tumor growth in $Ogt^{-/-}$ tumor-bearing mice, compared to the isotype antibody treatment group, and their antitumor activity of *Ogt* deficiency disappeared, both in tumor volume, weight and survival curve (*Figure 6J–K*). However, depletion CD4$^+$ T cells have no obvious effect both in tumor volume and weight in $Ogt^{-/-}$ tumor-bearing mice, compared to

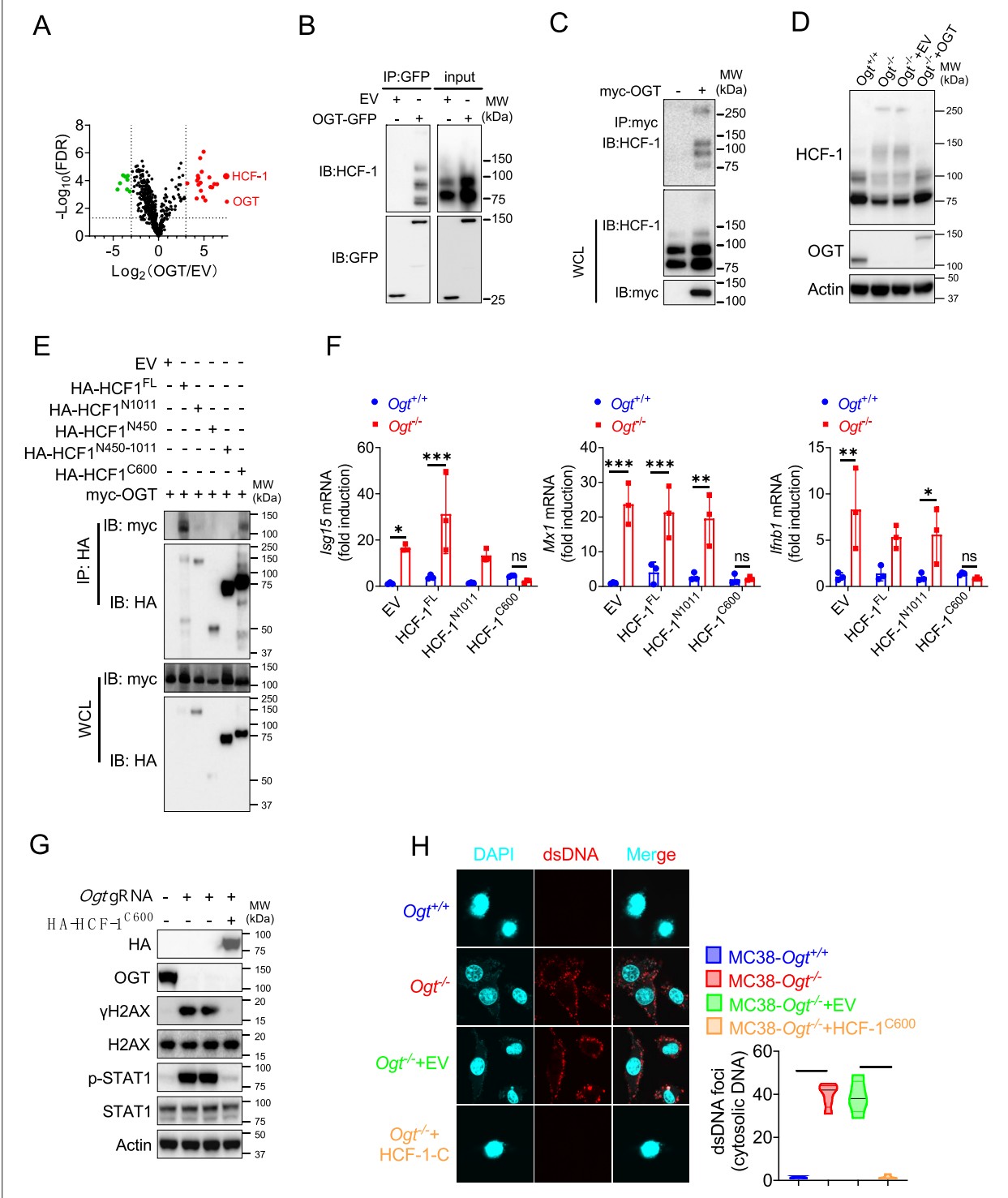

**Figure 5.** The C terminal of HCF-1 rescue DNA damage and the type I IFN pathway in *Ogt⁻/⁻* cells. (**A**) Volcano plot of OGT binding proteins identified by LC–MS/MS from stably expressed exogenous GFP-OGT in OGT knockout HT29 cells. (**B**) OGT and HCF1 binding was confirmed by immunoprecipitation assay in OGT rescued HT29 cells. (**C**) OGT and HCF1 binding was confirmed by immunoprecipitation assay in 293T cells. (**D**) HCF1 cleavage was confirmed by western blot in *Ogt* rescued MC38 cells. (**E**) Co-IP analysis of the interaction between OGT and different HCF-1 mutant. (**F**) Real-time PCR analysis of cytokines mRNA expression effected by HCF-1 isoforms in MC38 OGT knockout cells. (**G**) Western blot analysis of γH2AX and H2AX expression in exogenous HCF-1^C600 expressed MC38 *Ogt* knockout cells. (**H**) The extranuclear dsDNA were determined by anti-dsDNA fluorescence staining assay and was quantified by image J in exogenous HCF-1^C600 expressed MC38 OGT knockout cells. Data are representative of

*Figure 5 continued on next page*

*Figure 5 continued*

three or four independent experiments. Statistical significance was determined by one-way ANOVA, two-way ANOVA, *p<0.05, **p<0.01, ***p<0.001, ns, no significant difference. Data represent the mean of ± SD.

The online version of this article includes the following figure supplement(s) for figure 5:

**Figure supplement 1.** In vitro pull-down assay analysis of the interaction of HCF-1[C600] and OGT.

the isotype antibody treatment group (*Figure 6—figure supplement 3A*). These results implied that CD8[+] T cells mediated the inhibitory effect of *Ogt* deficiency on tumor progression.

Next, to determine whether this effect is specific for OGT knockout, we performed this experiment in vivo using MC38 tumor-bearing rescue model. As we expected, decreased tumor growth and prolong mice survival phenotype disappeared both in OGT rescued cells (*Figure 6L–M*) and in OGT/cGAS or OGT/STING double knockout tumors (*Figure 6N–O*). Furthermore, flow cytometry results showed that the proportion of CD4[+] T and CD8[+] T cells, CD8[+] IFN-γ[+] and CD8[+] TNF-α[+] disappeared both in OGT rescued cells (*Figure 6—figure supplement 3B-D*) and in OGT/cGAS or OGT/STING double knockout tumors in vivo (*Figure 6P* and *Figure 6—figure supplement 3E-F*). These results proved that OGT deficiency induces the cGAS/STING and activates the type I IFN pathway.

Increasing evidences support that intratumoral infiltration of CD8[+] T cells dictates the response to immune checkpoint blockade (ICB) therapy and its efficacy on various cancers (*Yost et al., 2019*; *Patel and Minn, 2018*). Blocking PD-L1 can restore the anti-tumor immune function and enhance the antitumor immunity by promoting CD8-positive T-cell infiltration, which is widely used in clinical immunotherapy (*Brahmer et al., 2012*; *Topalian et al., 2012*). Because *Ogt* deficiency induced tumor cell-intrinsic immune response to recruit CD8[+] T cells into MC38, LLC and B16-OVA cells, we hypothesized that *Ogt* deficiency potentiated enhance the efficacy of PD-L1 blockade in vivo. To test this hypothesis, we carried out the combination treatment of *Ogt* knockout and neutralizing antibody (anti-PD-L1). MC38 and LLC tumor growth was significantly delayed in tumor-bearing mice treated with PD-L1 antibody compared to isotype control (*Figure 6Q* and *Figure 6—figure supplement 3G*), which translated into extended survival (*Figure 6R* and *Figure 6—figure supplement 3H*). Deletion of *Ogt* synergized with PD-L1 blockade treatments to improve antitumor immunity. We next assessed the potential relevance of *OGT* in human cancer immunity. We first analyzed gene expression profiles of cancer patients from TCGA database and survival, we found that low *OGT* expression was associated with improved overall survival (OS) and progression free survival (PFS) in patients with COAD (*Figure 6S, T*). Using TIMER2.0 (http://timer.cistrome.org) analysis, we found that *OGT* expression negatively correlated with CD8[+] T cell infiltration (*Figure 6U*). For further analysis, we found that a set of genes associated with immune response was robustly regulated in *OGT* high and *OGT* low patients. The response interferon-gamma, interferon-gamma production, cellular defense response, regulation of inflammatory response, acute inflammatory response is upregulated; DNA mismatch repair is a downregulated process, as shown by Gene Ontology (GO) enrichment and pathway analysis (*Figure 6—figure supplement 4A*). Gene Set Enrichment Analysis (GSEA) showed that T cell activation, response to interferon-gamma, interferon-gamma production, antigen processing and presentation, interleukin-1/12 production, dectin-1-mediated noncanonical NF-κB signaling are negatively correlated with *OGT* expression (*Figure 6—figure supplement 4B-H*), while mismatch repair, covalent chromatin modification and DNA repair complex are positively correlated with *OGT* expression (*Figure 6—figure supplement 4I-K*). Of our most interest, we found that *CD8A*, *IFNG*, *ISG15*, *MX1*, *CD274*, and *CXCL10* expression are negatively correlated with *OGT* expression (*Figure 6—figure supplement 4L-Q*f). These data suggest a potential involvement of *OGT* deficiency and antitumor immunity in patients with cancer.

## Combination therapy with OSMI-1 and anti-PD-L1 Ab augmented T cells and antitumor immunity

Based on previous studies, we know that *Ogt* knockout causes DNA damage, accumulates cytosolic DNA, induces cGAS-STING pathway and activates antitumor immunity. Here, we speculated that OGT inhibitor may cause DNA damage and activate antitumor immunity. OSMI-1 is a small molecule inhibitor of OGT that does not significantly affect other glycosyltransferases and is active in a very low doses (*Ortiz-Meoz et al., 2015*). The cell proliferation exhibited no obvious difference in different

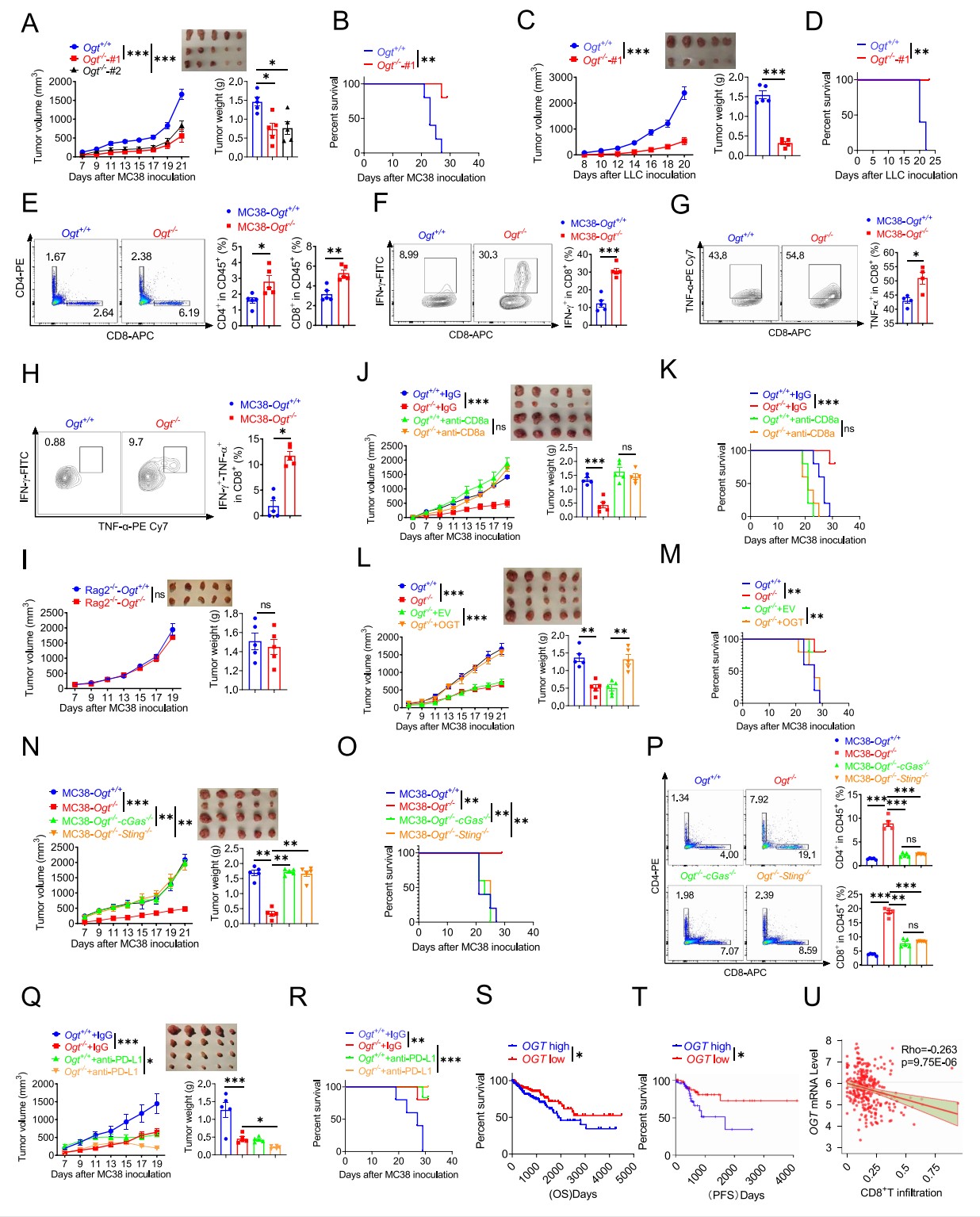

**Figure 6.** *Ogt* deficiency inhibits tumor progression through enhancing infiltration by CD8$^+$ T cells. (**A–B**) Tumor volume, weight of *Ogt*$^{+/+}$ or *Ogt*$^{-/-}$ MC38 tumors in C57BL/6 J mice and mice survival, n=5 respectively. (**C–D**) Tumor volume, weight of *Ogt*$^{+/+}$ or *Ogt*$^{-/-}$ LLC tumors in C57BL/6 J mice and mice survival, n=5 respectively. (**E–H**) Flow cytometry analysis of percentage of CD4$^+$ and CD8$^+$ T cells population (**E**) and IFN-γ$^+$ (**F**), TNF-α$^+$ (**G**), IFN-γ$^+$TNF-α$^+$ double positive (**H**) intratumoral CD8$^+$ T cells population in MC38 tumors, subcutaneous tumor isolated at day 18 post-tumor inoculation, n=5 respectively. (**I**) Tumor volume and weight of *Ogt*$^{+/+}$ or *Ogt*$^{-/-}$ MC38 in *Rag2*$^{-/-}$ mice, n=5 respectively. (**J–K**) Tumor volume, weight of *Ogt*$^{+/+}$ or *Ogt*$^{-/-}$ MC38 tumors injected with either control IgG or anti-CD8α at days 0, 7, and 14 post tumor inoculation in C57BL/6 J mice and mice survival,

*Figure 6 continued on next page*

*Figure 6 continued*

n=5 respectively. (**L–M**) Tumor volume, weight of $Ogt^{-/-}$ rescued MC38 tumors in C57BL/6 J mice, tumor growth volume and weight (**L**), mice survival (**M**). **N–O** Tumor volume, weight of $Ogt^{-/-}cGAS^{-/-}$ or $Ogt^{-/-}Sting^{-/-}$ double knockout MC38 tumors in C57BL/6 J mice, tumor growth volume and weight (**N**), mice survival (**O**).(**P**) Flow cytometry analysis showing percentage of CD4$^+$ and CD8$^+$ T cells population (**N**), CD8$^+$ IFN-γ$^+$ (**O**), CD8$^+$ TNF-α$^+$ T cell population (**P**) in $Ogt^{-/-}cGAS^{-/-}$ or $Ogt^{-/-}Sting^{-/-}$ double knockout MC38 tumors in C57BL/6 J mice, subcutaneous tumor isolated at day 18 post-tumor inoculation. (**Q–R**) Tumor volume, weight of $Ogt^{+/+}$ or $Ogt^{-/-}$ MC38 tumors injected with either control IgG or anti-PD-L1 at days 7, 10, and 13 post tumor inoculation in C57BL/6 J mice and mice survival, n=5 respectively. (**S**) Kaplan-Meier survival curves for colorectal cancer patients with low (n=207) or high (n=231) *OGT* transcripts in TCGA dataset. (**T**) Progression-free survival curves for colorectal cancer patients with low (n=58) or high (n=58) *OGT* transcripts in TCGA dataset. (**U**) Scatterplot presenting the association between the mRNA expression level of *OGT* and CD8$^+$ T infiltration, Spearman's r=–0.263, P=9.75E-6, Spearman's rank correlation test. Data are representative of two or three independent experiments. Statistical significance was determined by Spearman's rank correlation test, unpaired Student's t-test, one-way ANOVA, two-way ANOVA, *p<0.05, **p<0.01, ***p<0.001, ns, no significant difference. Data represent the mean of ± SD.

The online version of this article includes the following figure supplement(s) for figure 6:

**Figure supplement 1.** The cell proliferation of different tumor model in vitro and B16-OVA tumor growth analysis in vivo.

**Figure supplement 2.** *Ogt* deficiency inhibits tumor progression through enhancing infiltration of CD8$^+$ T cells.

**Figure supplement 3.** *Ogt* deficiency inhibits tumor progression through enhancing infiltration of CD8$^+$ T cells.

**Figure supplement 4.** *OGT* expression is negatively related to CD8$^+$ T cells infiltration in human colorectal cancer.

concentrations in MC38 and LLC cells in vitro assay (*Figure 7—figure supplement 1A-B*). We stained cytosolic dsDNA with anti-dsDNA and found that treatment with OSMI-1 could significantly induce a high percentage of cytosolic DNA accumulation (*Figure 7A* and *Figure 7—figure supplement 1C*). We next examined the DNA damage and found that γH2AX was obviously increased in OSMI-1-treated cells (*Figure 7B–C*). We also performed the anti-γH2AX immunofluorescence staining and found that OSMI-1 significantly induced immense DNA strand breakage in treated cells (*Figure 7D* and *Figure 7—figure supplement 1D*).

As we all know, the presence of cytosolic DNA could trigger activation of cGAS-STING pathway (*Chen et al., 2016*). To investigate whether OSMI-1 activated cGAS/STING pathway, we examined activation of major regulators of the pathway in OSMI-1-treated MC38 and LLC cells, as indicated by increased phosphorylation of STAT1, TBK1, and IRF3 and reduced STING expression (*Figure 7B–C*), which is consistent with *Ogt* knockout cells. As mentioned earlier, deletion of *Ogt* synergized with PD-L1 blockade treatments to improve antitumor immunity. We next sought to determine whether OSMI-1 enhanced the antitumor immune effect of anti-PD-L1 antibody in vivo by using MC38 tumor-bearing model. To our surprise, similar to anti-PD-L1 therapy, OSMI-1 alone significantly inhibited MC38 tumor growth and survival, and the combination of OSMI-1 and anti-PD-L1 therapy resulted in superior tumor suppression compared with monotherapy (*Figure 7E–F*). Flow cytometry results showed that proportion of CD4$^+$ T and CD8$^+$ T cells was increased both in OSMI-1, anti-PD-L1 treatment alone and combined treatment with OSMI-1 and anti-PD-L1 antibody (*Figure 7I*), production of IFN-γ and TNF-α were significantly enhanced in intratumoral CD8$^+$ T cells not only combined treatment with OSMI-1 and anti-PD-L1 antibody, but also single-agent OSMI-1 treatment (*Figure 7J–K*). This pharmacological inhibition model is consistent with MC38 *Ogt* knockout tumor-bearing mice model.

Furthermore, we treated mice bearing LLC tumors, because it represents a most aggressive murine tumor model and are highly resistant to various immunotherapies. Our results showed that treatment with single-agent OSMI-1 can slightly inhibited LLC tumor growth and survival. Combined treatment with OSMI-1 and anti-PD-L1 antibody caused significantly greater tumor suppression than either monotherapy (*Figure 7F–G*). We also found that proportion of CD4$^+$ T and CD8$^+$ T cells was significantly increased in combined treatment with OSMI-1 and anti-PD-L1 antibody. However, single-agent OSMI-1 or anti-PD-L1 antibody treatment cannot significantly increased the proportion of CD4$^+$ T and CD8$^+$ T cells (*Figure 7L*). Production of IFN-γ and TNF-α were significantly enhanced in intratumoral CD8$^+$ T cells in combined treatment with OSMI-1 and anti-PD-L1 antibody, but not in single-agent OSMI-1 or anti-PD-L1 treatment (*Figure 7M–N*). This model is also consistent with *Ogt* knockout tumor-bearing mice model. In summary, OGT inhibitor OSMI-1 induces DNA damage and cytosolic DNA accumulation, activates cGAS/STING pathway. Combined OGT inhibitor with anti-PD-L1 antibody markedly suppressed tumor growth and increased CD8$^+$ T cells and production of IFN-γ and TNF-α in tumor. These results demonstrated a pivotal role of OGT inhibition in augmenting

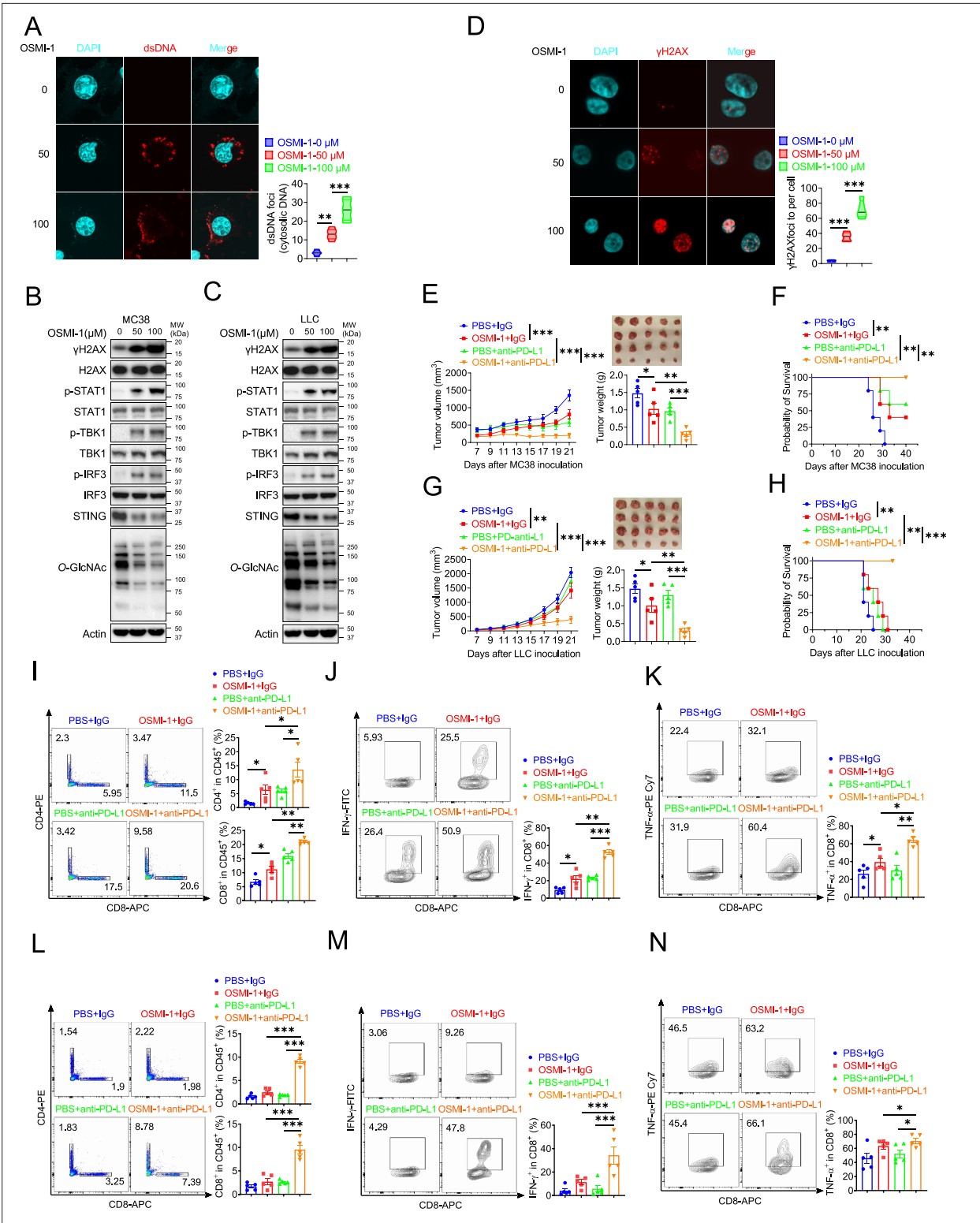

**Figure 7.** Combination therapy with OSMI-1 and anti-PD-L1 augmented T cells and antitumor immunity. (**A**) The extranuclear dsDNA were determined by anti-dsDNA fluorescence staining treated with 50 μM and 100 μM OSMI in MC38 cells respectively and was quantified by image J. (**B–C**) Western blot analysis of protein expression in MC38 and LLC cells treated with 50 μM and 100 μM OSMI, respectively. (**D**) Analysis of γH2AX and H2AX expression by anti-γH2AX staining treated with 50 μM and 100 μM OSMI in MC38 cells and was quantified by image J. (**E–F**) Tumor volume, weight of MC38 tumors injected with either control OSMI-1 or anti-PD-L1 in C57BL/6 J mice and mice survival. (**G–H**) Tumor volume, weight of LLC tumors injected with either control OSMI-1 or anti-PD-L1 in C57BL/6 J mice and mie survival. (**I–K**) Flow cytometry analysis showing percentage of CD4[+] and CD8[+] T cells

*Figure 7 continued on next page*

*Figure 7 continued*

population (**I**), and CD8$^+$ IFN-γ$^+$ cells (**J**), CD8$^+$ TNF-α$^+$ cells (**K**) population in MC38 subcutaneous tumor isolated at day 18 post-tumor inoculation. (**L–N**) Flow cytometry analysis showing percentage of CD4$^+$ and CD8$^+$ T cells population (**L**), and CD8$^+$ IFN-γ$^+$ cells (**M**), CD8$^+$ TNF-α$^+$ cells (**N**) population in LLC subcutaneous tumor isolated at day 18 post-tumor inoculation. Data are representative of three or four independent experiments. Statistical significance was determined by unpaired Student's t-test, one-way ANOVA, two-way ANOVA, *p<0.05, **p<0.01, ***p<0.001, ns, no significant difference. Data represent the mean of ± SD.

The online version of this article includes the following figure supplement(s) for figure 7:

**Figure supplement 1.** OSMI-1 could significantly induce a high percentage of DNA damage.

the antitumor immune response of ICB. Given the increasing importance of immunotherapy for the management of patients with that OGT inhibitors, combined with anti-PD-L1 blockade may offer a particularly attractive strategy for the treatment of colorectal and lung cancer, which are instrumental in turning 'cold tumors' into 'hot tumors'.

## Discussion

As an important metabolic enzyme, OGT promotes tumorigenesis by glycosylating numerous proteins. This study found that OGT levels are elevated in both human and mouse tumors (*Figure 1*). Additionally, epithelial deletion of OGT inhibits colorectal tumorigenesis in mice (*Figure 2*). DNA damage and DNA repair signaling pathways play pivotal roles in maintaining genomic stability and integrity by correcting impaired DNA, which otherwise can contribute to carcinogenesis (*Clementi et al., 2020*). These pathways are activated in response to endogenous or exogenous DNA-damaging agents, helping cells to preserve genomic stability. HCF1 is a member of the host cell factor family, involved in regulating the cell cycle and playing regulatory roles in various transcription-related processes. This study found that OGT deficiency leads to DNA damage and cytosolic DNA accumulation (*Figure 4*) by regulating HCF-1 cleavage and maturation (*Figure 5*). In antitumor therapies, chemotherapy and radiotherapy induce cell death by directly or indirectly causing DNA damage, thereby increasing tumor sensitivity to cancer therapies. DNA damage-inducing therapies have proven to be immensely beneficial for cancer treatment, functioning by directly or indirectly forming DNA lesions and subsequently inhibiting cellular proliferation. Therefore, targeting DNA repair pathways may represent a promising therapeutic approach for cancer treatment. In this study, we observed that deletion of OGT leads to uncontrolled expansion of DNA damage and induces cytosolic dsDNA accumulation in tumor cells, suggesting that OGT is a promising therapeutic target for cancer treatment.

The cytosolic DNA sensing pathway has emerged as the major link between DNA damage and innate immunity, DNA damage in the nucleus results in the accumulation of cytosolic DNA, which activate the cGAS–STING pathway (*Lan et al., 2014*; *Härtlova et al., 2015*; *Yang et al., 2017*; *Glück et al., 2017*; *Harding et al., 2017*; *Mackenzie et al., 2017*). This study found that OGT deficiency activates the cGAS/STING-dependent IFN-I pathway (*Figure 3*), resulting in the expression of proinflammatory cytokines (e.g. IFNB1 and ISGs) and chemokines (e.g. CXCL10) in a TBK1-IRF3-dependent manner. OGT deficiency causes DNA damage and cytosolic DNA accumulation, which triggers this response. The cGAS–STING pathway is the key cytosolic DNA sensor responsible for the type I IFN production, DC activation, and subsequent priming of CD8$^+$ T cells against tumor-associated antigens (*Woo et al., 2014*; *Li et al., 2013*; *Deng et al., 2014*). Recent evidence shows that proper activation of tumor cell-intrinsic immunity or innate immune cells can enhance antitumor immunity (*Lu et al., 2021*; *Guan et al., 2021*; *Zhang et al., 2021*). In this study, cGAS-STING-IIFNB1-CXCLl10 signaling axis can provide an activated signal for epitope-specific CD8$^+$ T cell priming by antigen-presenting cells. Ultimately, this process results in increased infiltration of tumor-infiltrating CD8 +T lymphocytes and more effective inhibition of tumor growth within the tumor microenvironment (TME) (*Figure 6*). However, knockout of cGAS or STING can eliminate the proliferation signal of CD8 +T cells and abolish antitumor immunity.

Immunotherapy with checkpoint-blocking antibodies targeting CTLA-4 and PD-1/PD-L1 has revolutionized cancer treatment and drastically improved the survival of individuals in the clinical treatment. Although immunotherapy has made great progress in the treatment of solid tumors, only around 20% of patients with non-small cell lung cancer (NSCLC) respond to mono-immunotherapy, and a large proportion of individuals develop resistance. Therefore, there is a need to explore novel alternative

strategies and personalized immunotherapy strategies through combinations of PD-1/PD-L1 blockade with small molecular targets. This is aimed at improving sensitivity to activated antitumor immune responses in patients and addressing drug resistance (*Jenkins et al., 2018*; *Murciano-Goroff et al., 2020*). Here, we showed that the OGT inhibitor OSMI-1 induced DNA damage and cytosolic DNA accumulation which led to activation of the cGAS-STING-TBK1-IRF3 pathway, then enhanced the innate and adaptive immune responses to tumor cells, which reversed the immunosuppressive TME by increasing CD8[+] T cells infiltration. To our surprise, especially in lung cancer mice model, combination therapy with OSMI-1/PD-L1 can achieve a better antitumor effect than either monotherapy (*Figure 7*).

In summary, our findings demonstrated deficiency in OGT mediated genomic instability and result in cytosolic dsDNA accumulation, which activating the cGAS-STING signaling pathway, increasing inflammatory cytokines, and enhancing antitumor immunity. Our study also addresses an unmet clinical need through the combination of OGT inhibition and anti-PD-L1 therapy, which may represent a promising strategy for colorectal and lung cancer therapy.

## Materials and methods

### Mice

*Ogt* IEC cKO *Apc^{min}* mice were generated by crossing the *Ogt^{fl/fl}* mice with Villin-Cre mice, then crossed with *Apc^{min}* mice. C57BL/6 mice, *Apc^{min}* mice, Villin-Cre mice, *Rag2^{-/-}* mice, *Ifnar1^{-/-}* mice and OT-I mice were purchased from Jackson Laboratories. Mice between 8–10 weeks of age were used for the animal experiments, tail genomic DNA was isolated for genotyping. Primers for genotyping PCR are listed in *Supplementary file 1a*. All in vivo experiments were conducted in accordance with the National Institutes of Health Guide for the Care and Use of Laboratory Animals and the Institutional Animal Care and Use Committee. The study was approved by the Ethics Committee of The Ohio State University and all procedures were conducted in accordance with the experimental animal guidelines of The Ohio State University (Project ID 2018A00000022). Institutional Biosafety Committee (IBC) Protocol: 2018R00000017-R1.

### Cell lines and plasmids

Cell lines used in this study including B16-OVA cell line (murine melanoma, SCC-420) from Sigma-Aldrich, 293T cell line (CRL-3216), MC38 (RRID: CVCL_B288), LLC cell line (CRL-1642) and HT29 cell line (HTB-38) from the American Type Culture Collection. The identity of all cell lines has been authenticated with STR profiling by provider. All the cell lines tested negative for mycoplasma contamination. We have not used any cell lines from the list of commonly misidentified cell lines maintained by the International Cell Line Authentication Committee. 293T, MC38, B16-OVA, LLC and HT29 cells were cultured in Dulbecco's Modified Eagle Medium (Gibco) supplemented with 10% fetal bovine serum (Millipore Sigma), 1% glutamine (Gibco), 1% sodium pyruvate, 1% non-essential amino acids (Gibco), 100 IU/ml penicillin and 100 mg/ml streptomycin (Gibco). The pWPXLd-OGT-GFP fusion vector was described in our previous articles,[2] pET24a-ncOGT-FL (190821, Addgene), other plasmids were cloned into pcDNA3.1 backbone with c-myc or HA tag.

### Quantitative real-time PCR

Total RNA was extracted from in vitro cultured cells and tissues using Trizol reagent (Invitrogen). cDNA synthesis was performed with Moloney murine leukemia virus reverse transcriptase (Invitrogen) at 38 °C for 60 min. RT-PCR was performed using iTaq Universal SYBR Green Supermix in CFX Connect Real-Time PCR Detection System. The fold difference in mRNA expression between treatment groups was determined by a standard $^{\triangle\triangle}$Ct method. *β-actin and GAPDH* were analyzed as an internal control. The primer sequences of individual genes are listed in the *Supplementary file 1b*.

### Co-immunoprecipitation (Co-IP) and western blot

For co-immunoprecipitation, cells were lysed in RIPA buffer supplemented with Protease Inhibitor Cocktail. Total protein extracts were incubated with goat anti-GFP Trap agarose (gta-20, Chromotek) or anti-c-Myc Agarose (20168, Thermo Fisher Scientific) overnight at 4 °C under gentle agitation. Samples were washed four times with cold RIPA buffer. To elute proteins from the beads, samples

were incubated with 50 μl of SDS sample buffer at 95 °C for 10 min. Protein content in the supernatant was analyzed by western blot. For western blot, electrophoresis of proteins was performed by using the NuPAGE system (Invitrogen) according to the manufacturer's protocol. Briefly, cultured cells were collected and lysed with RIPA buffer. Proteins were separated on a NuPAGE gel and were transferred onto nitrocellulose membranes (Bio-Rad). Appropriate primary antibodies and HRP-conjugated secondary antibodies were used and proteins were detected using the Enhanced Chemiluminescent (ECL) reagent (Thermo Fisher Scientific). The images were acquired with ChemiDoc MP System (Bio-Rad). Primary antibodies for western blot included anti-OGT (5368, Cell Signaling Technology), anti-$O$-GlcNAc (ab2739, Abcam), anti-MAVS (sc-365334, Santa Cruz Biotechnology), anti-phospho-TBK1/NAK (Ser172) (5483, Cell Signaling Technology), anti-TBK1 (3504, Cell Signaling Technology), anti-phospho-IRF3 (Ser396) (4947, Cell Signaling Technology), anti-IRF3 4962, Cell Signaling Technology, anti-anti-phospho-STAT1 (Ser727) (8826 S, Cell Signaling Technology), anti-STAT1 (9172, Cell Signaling Technology), anti-STING (D2P2F) (13647, Cell Signaling Technology), anti-cGAS for mouse (31659 S, Cell Signaling Technology), anti-cGAS for human (15102 S, Cell Signaling Technology), anti-Phospho-Histone H2AX (Ser139) (MA1-2022, Thermo Fisher Scientific), anti-H2AX (2595 S, Cell Signaling Technology), anti-HCFC1 (50708 S, Cell Signaling Technology), anti-GFP (sc-9996, Santa Cruz Biotechnology), anti-Actin (sc-1615, Santa Cruz Biotechnology), anti-Myc-peroxidase (11814150001, Millipore Sigma) and anti-HA-HRP (26183, Thermo Fisher Scientific).

### In vitro pull-down assay

Recombinant OGT-his purified from *E. coli* system. The recombinant HA-HCF-1$^{C600}$ were expressed in 293T cells and total protein extracts were incubated with Pierce Anti-HA Magnetic Beads (88836, Thermo Fisher Scientific) overnight at 4 °C under gentle agitation, then competitively with HA Synthetic Peptide (PP100028, Sino Biological) to purify HA-HCF-1$^{C600}$ recombinant protein. Proteins were then mixed with the Ni$^{2+}$ beads for 2 hr with rotation at 4 °C, The beads were then washed with washed four times with cold RIPA buffer, and protein samples were analyzed by western blot and commassie blue staining.

### ELISA

Cytokines generated by in vitro cultured tissues from mice were quantified using the ELISA Set for mouse IL-2, IL-6, IL-10, IL-12a, IL-17, IFN-α, IFN-β, IFN-γ, CXCL10 and TNF-α (BD Biosciences) according to the manufacturer's protocol.

### Lentivirus-medicated gene knockout in mice and human cell lines

pLenti-CRISPR-V2 vector was used for CRISPR/Cas9-mediated gene knockout in MC38, B16-OVA, LLC and HT29 cell lines, all primers used for sgRNA are listed in the *Supplementary file 1c*. Briefly, lentivirus vector expressing gRNA was transfected together with package vectors into 293T package cells. 48 h rand 72 hr after transfection, virus supernatants were harvested and filtrated with 0.2 μm filter. Target cells were infected twice and 2 μg/mL puromycin was added at 3–5 days for selection. After that, the positive cells were diluted into 96-well plates at one cell per well. Isolated single clones were verified by western blot.

### Quantification of cytosolic DNA

For PicoGreen staining, cells were washed twice with cold PBS and fixed with cold methanol at –20 °C for 10 min. After being washed three times with PBS, cells were blocked with 1% BSA in PBS for 1 hr and stained with Pico488 dsDNA quantification reagent for 1 hr. After being washed three times with PBS, the dish was mounted on white microscope slides using the Prolong Gold Antifade Mountant regent with DAPI and imaged on confocal microscope.

For anti-dsDNA staining, cells were washed with 1×PBS. Fix the cells with fresh 4% of paraformaldehyde (sc-281692, Santa Cruz Biotechnology) for 10 min at room temperature, then discard the 4% PFA in an appropriate container, wash the cells with 1×PBS, incubate the cells with the permeabilization buffer for 7 min at room temperature. After three additional washes with 1×PBS, block nonspecific binding sites by incubating the cells with the blocking buffer for 30 min at room temperature. Remove the blocking buffer (Do not wash). Anti-dsDNA antibody (sc-58749, Santa Cruz Biotechnology) at 1:100 in 1% BSA-PBST.

Incubate samples with diluted anti-dsDNA antibody in humidified chamber overnight at 4 °C. Wash three times by 1×PBS. Dilute the secondary antibody (goat anti-mouse IgG H&L Alexa Fluor 488 preabsorbed, ab150117, Abcam) at 1:200 in 1% BSA-PBST. Incubate samples with the diluted secondary antibody for 1 hr at room temperature. Wash 3 times by 1×PBS (for 5 min at room temperature). Drop mounting media containing DAPI (Vector laboratories, Vectashield Hardset Anti-fade mounting medium with DAPI, H-1500) on a slide, and put carefully the cover slip. Let it sit for 1–3 hr at room temperature. Keep the slide overnight at 4 °C in slide box. Observe and acquire pictures with a fluorescence microscope using the RFP and DAPI channels the next day to ensure that the mounting medium is completely dry.

## Immunohistochemistry (IHC) and immunofluorescence staining

For immunostaining of tissue sections, 5 µm paraffin-embedded sections were cut from paraffin blocks of biopsies. Tissue slides were placed in oven at 60 °C for half 1 hr and then deparaffinized in xylene three times for 5 min each followed by dipping in graded alcohols (100%, 95%, 80%, and 70%) three times for 2 min each. Slides were washed with distilled water (dH$_2$O) 3 times for 5 min each and immersed in 3% hydrogen peroxide for 10 min followed by washing with dH$_2$O. Slides were transferred into pre-heated 0.01 M Citrate buffer (pH 6.0) in a steamer for 30 min, and then washed with dH$_2$O and PBS after cooling. Slides were blocked with 3% BSA/PBS at room temperature for 1 hr and then incubated with primary antibody overnight at 4 °C, followed by incubating with secondary antibody including Biotinylated Anti-rabbit IgG and Biotinylated Anti-mouse IgG at room temperature for 1 hr. After incubation with avidin-biotin complex followed by washing 3×5 min with PBS, slides were washed with tap water, counterstained with hematoxylin and dipped briefly in graded alcohols (70%, 80%, 95%, and 100%) in xylene two times for 5 min each. Finally, slides were mounted and imaged by confocal microscopy. For the histological scoring, image J software was used and scored in a blinded fashion using a previously published paper (*Young and Morrison, 2018*).

For the immunofluorescence staining on cells, cells were cultured on the dish. After treatment, cells were washed with PBS, and then fixed with 4% paraformaldehyde (PFA) for 10 min, and permeabilized by 0.1% triton X-100. Non-specific binding was blocked through incubation with 5% BSA for 1 hr. Cells were stained with anti-γH2AX (05–636-I, Millipore Sigma) overnight at 4 °C, and then incubated with fluorochrome-conjugated 2 nd antibodies for 1 hr at room temperature. Nucleus was visualized by mounting with DAPI-containing. Finally, cells were imaged by confocal microscopy.

For the histological scoring, slides were then examined and scored in a blinded fashion using a previously published grading system (*Takagi et al., 2011*). Briefly, histology was scored as follows:– Epithelium (E): 0, normal morphology; 1, loss of goblet cells; 2, loss of goblet cells in large areas; 3, loss of crypts; and 4, loss of crypts in large areas. –Infiltration (I): 0, no infiltration; 1, infiltration around crypt bases; 2, infiltration reaching the muscularis mucosa; 3, extensive infiltration reaching the muscularis mucosa and thickening of the mucosa with abundant edema; and 4, infiltration of the submucosa. The total histological score was the sum of the epithelium and infiltration scores (total score = E + I), and thus ranged from 0 to 8.

## Comet SCGE assays

Cells were trypsinized to a single cell suspension. Dilute approximat 1:1 in PBS, and immediate place 1 ml of cell suspension in a 1.5 ml tube on mice. Count cells and ensure a density between 10$^6$ /ml suspension, add 5 µl cell suspension and 50 µl melted LMAgarose. Mix well and drop it on the slide at 37 °C. Place the slides immediately at 4 °C for 30 min. Drop slides immersed in cold lysis solution at 4 °C for 30 min. After cell lysis, electrophoresis was then carried out in the TBE for 30 min at 35 V voltage. Lastly, DNA was stained with EtBr (20 mg/ml) dye for 10 min. Slides were completely air-dried before taking images. Images were taken by the confocal microscopy and analyzed by using Comet-Score 2.0.

## In vitro cross-priming of T cells by BMDCs

BMDCs were prepared by flushing bone marrow from mouse hindlimbs and plating 1×10$^6$ cells/ml in RPMI 1640 media with 10% FBS and 20 ng/ml mGM-CSF. Fresh medium with mGM-CSF was added into the wells on day 4. On day 6, immature BMDCs were harvested and loaded with 1 µg/ml OVA[257-264] (GenScript), B16-OVA-*Ogt*[+/+] and B16-OVA-*Ogt*[−/−] cells supernatant at 37 °C for 6 h. BMDCs were

then washed three times with PBS to remove excessive peptide followed by resuspension in RPMI 160 medium with 10% FBS. OT-I CD8[+] T cells were harvested from spleens of wildtype OT-1 mice by CD8[+] T Cell Enrichment Kit (Miltenyi), labeled with CFSE with 5 µM CFSE (carboxyfluorescein succinimidyl ester, Life Technologies) in PBS containing 0.1% BSA (Millipore Sigma) for 8 min at 37 °C. CFSE-labeled OT-I CD8[+] T cells were co-cultured with OVA$_{257-254}$ peptide, B16-OVA-*Ogt$^{+/+}$* and B16-OVA-*Ogt$^{-/-}$* cells supernatant pulsed BMDCs at a 5:1 ratio in 96-well plates. Analysis of the in vitro expansion was performed 48 hr after co-culture by enumerating the number of CFSE-diluted CD8[+] T cells.

## Mass spectrometry assay of OGT interactome

High resolution/accurate mass-based quantitative proteomics strategy was employed to identify protein-protein interactions. Briefly, immunoprecipitated (GFP) OGT complex from *Ogt$^{-/-}$*+GFP and *Ogt$^{-/-}$*+OGT GFP in HT29 cells were boiled with SDS buffer followed by E3filter digestion (*Martin et al., 2024*). The digests were desalted using C18 StageTips, dried in a SpeedVac and then resuspended in 20 µl LC buffer A (0.1% formic acid in water) for LC-MS/MS analysis. The analysis was performed using an Orbitrap Eclipse MS (Thermo Fisher Scientific) coupled with an Ultimate 3000 nanoLC system and a nanospray Flex ion source (Thermo Fisher Scientific). Peptides were first loaded onto a trap column (PepMap C18; 2 cm ×100 µm I.D.) and then separated by an analytical column (PepMap C18, 3.0 µm; 20 cm ×75 mm I.D.) using a binary buffer system (buffer A, 0.1% formic acid in water; buffer B, 0.1% formic acid in acetonitrile) with a 165 min gradient (1%–25% buffer B over 115 min; 25% to 80% buffer B over 10 min; back to 2% B in 5 min for equilibration after staying on 80% B for 15 min). MS data were acquired in a data-dependent top-12 method with a maximum injection time of 20ms, a scan range of 350–1800 Da, and an automatic gain control target of 1e6. MS/MS was performed via higher energy collisional dissociation fragmentation with a target value of 5e$^5$ and maximum injection time of 100ms. Full MS and MS/MS scans were acquired by Orbitrap at resolutions of 60,000 and 17,500, respectively. Dynamic exclusion was set to 20 s. Protein identification and quantitation were performed using the MaxQuant-Andromeda software suite (version 1.6.3.4) with most of the default parameters. Other parameters include: trypsin as an enzyme with maximally two missed cleavage sites; protein N-terminal acetylation and methionine oxidation as variable modifications; cysteine carbamidomethylation as a fixed modification; peptide length must be at least seven amino acids. False discovery rate was set at 1% for both proteins and peptides.

## Colitis-associated carcinogenesis (CAC) animal model

The induction of AOM +DSS tumorigenesis model, mice received a single intraperitoneal injection (10 mg/kg body weight) of AOM followed by three cycles of 2.5% DSS exposure for 5 days. Mice were sacrificed and tumor assessments were made 8 weeks after AOM injection. Body weight and tumor number were measured for each animal at the completion of each study. Finally, the colon tissues were collected for further study.

## Tumor cell inoculation

For tumor growth, 5×10$^5$ MC38, LLC and B16-OVA cells were inoculated subcutaneously in the right flank at C57BL/6 mice or *Rag2$^{-/-}$* mice. For CD8[+] and CD4[+] T cell depletion, mice were treated with 200 µg of control IgG (clone LTF-2, Bio X cell) or anti-CD8α depleting antibody (clone 2.43, Bio X cell) at days 0, 7, and 14 post tumor cell inoculation. For PD-L1 blockade, mice were intraperitoneally injected with 250 µg of control IgG or anti-PD-L1 antibody (clone 10 F.9G2, Bio X Cell) at day 7, 10 and 13 post tumor cell inoculation. For OSMI-1 treatment experiment, OSMI-1 (10 mg/kg; Aobious, AOB5700) was administered intraperitoneally every two days from 3 to 19 days after post tumor cell inoculation, Digital caliper was used to measure tumor volume at least three times a week and tumor volume were calculated using the formula mm$^3$ = (Length ×width × width/2). Mice were sacrificed at 18 days for flow cytometry and sacrificed when tumors reached a size of 2000 mm$^3$ for survival curve.

## Flow cytometry for TME

Mice tumors were dissected and weighed, then minced into small fragments and digested with 1 mg/mL collagenase IV and 50 U/mL DNase I for 30 min at 37 °C. The cell suspensions were mechanically disaggregated and filtered with 100 µm cell strainers. Centrifuge and lysed with the

ammonium-chloride-potassium (ACK) lysis buffer for 5 min, then added PBS and passed through 100 μm cell strainers. Single cell suspensions were treated with purified anti-CD16/32 (Fc receptor block, clone 93; BioLegend), and then stained with fluorochrome-conjugated monoclonal antibodies, including anti-CD11b-FITC (M1/70), anti-F4/80-APC (BM8), anti-CD11c-PE-Cy5 (N418), anti-Ly6C-PE (HK1.4), anti-CD4-PE (GK1.5), anti-CD8-APC (53–6.7), anti-CD25-PE-Cy5 (PC61) from BioLegend. For intracellular cytokine staining of tumor-infiltrating lymphocytes (TILs), cells were stimulated in vitro with phorbol-12-myristate 13-acetate (PMA) (50 ng/ml, Millipore Sigma) and ionomycin (500 ng/ml, Millipore Sigma) in the presence of GolgiPlug and GolgiStop (BD Biosciences) for 4 hr, and then surface stained as aforementioned. Cells were then fixed and permeabilized using BD Cytofix/Cytoperm (BD Biosciences) and stained with anti-IFN-γ (XMG1.2) and anti-TNF-α (MP6-XT22) from BioLegend. For intranuclear Foxp3 staining, single-cell suspensions were stained with antibodies against cell-surface antigens as aforementioned, fixed and permeabilized using Foxp3 Fix/Perm Buffer Kit (BioLegend) followed by staining with Foxp3 (clone MF-14; BioLegend).

## Statistics analysis

Data were analyzed on GraphPad Prism 8 (GraphPad Software) and R software v4.2.2. The statistical tests, replicate experiments and p values are all indicated in the figures and/or legends. p Values were calculated using two-tailed student's *t* test, one-way ANOVA or two-way ANOVA with Tukey's multiple comparisons test, pearson correlation test, log-rank (Mantel-Cox) test for Kaplan-Meier survival analysis, two-way ANOVA with Sidak's multiple comparisons test or Tukey's multiple comparisons test, Hypergeometric test and adjusted with Benjamini-Hochberg method correction and two-sided Wilcoxon's rank-sum test and adjusted with Bonferroni's correction. Differences between groups are shown as the mean ± SD.

## Acknowledgements

We thank members of the Wen lab for discussion. This work was supported by National Institutes of Health (NIH) grants R01GM135234 and R01AI162779 (HW), R01AI077283 and R01CA262089 (ZL), R01DE026728 (YL) and The Ohio State University Comprehensive Cancer Center Intramural Research Program (AM, DMJ, KH and HW).

## Additional information

### Funding

| Funder | Grant reference number | Author |
|---|---|---|
| National Institutes of Health | R01GM135234 | Haitao Wen |
| National Institutes of Health | R01AI162779 | Haitao Wen |
| National Institutes of Health | R01AI077283 | Zihai Li |
| National Institutes of Health | R01CA262089 | Zihai Li |
| National Institutes of Health | R01DE026728 | Yu L Lei |
| The Ohio State University Comprehensive Cancer Center | Intramural Research Program | Haitao Wen |

The funders had no role in study design, data collection and interpretation, or the decision to submit the work for publication.

## Author contributions

Jianwen Chen, Bao Zhao, Hong Dong, Data curation, Formal analysis, Writing – original draft, Project administration, Writing – review and editing; Tianliang Li, Data curation, Formal analysis; Xiang Cheng, Wang Gong, Jing Wang, Junran Zhang, Gang Xin, Yu L Lei, Jennifer D Black, Zihai Li, Formal analysis; Yanbao Yu, Data curation; Haitao Wen, Conceptualization, Supervision, Funding acquisition, Writing – original draft, Writing – review and editing

## Author ORCIDs

Hong Dong  https://orcid.org/0000-0001-6719-4880
Haitao Wen  https://orcid.org/0000-0002-6363-2748

## Ethics

Ethics approval and consent to participate The National Institutes of Health Guide for the Care and Use of Laboratory Animals was followed in this study. The study was approved by the Ethics Committee of The Ohio State University and all procedures were conducted in accordance with the experimental animal guidelines of The Ohio State University (Project ID 2018A00000022). Institutional Biosafety Committee (IBC) Protocol: 2018R00000017-R1.

Reviewer #1 (Public Review): https://doi.org/10.7554/eLife.94849.3.sa1
Reviewer #2 (Public Review): https://doi.org/10.7554/eLife.94849.3.sa2
Author response https://doi.org/10.7554/eLife.94849.3.sa3

## Additional files

### Supplementary files

• Supplementary file 1. Primer sequences for genotype, RT-PCR and molecular cloning. (a) Primer sequences for genotype. (b) Related to Experimental Procedures. Primer sequences for RT-PCR. (c) Related to CRISPR/Cas9. Primer sequences for molecular cloning.

• Supplementary file 2. Mass spectrometry assay of OGT interactome.

• MDAR checklist

### Data availability

All data generated or analysed during this study are included in the manuscript and supporting files; source data files have been provided at Dryad https://doi.org/10.5061/dryad.4mw6m90ks.

The following dataset was generated:

| Author(s) | Year | Dataset title | Dataset URL | Database and Identifier |
|---|---|---|---|---|
| Chen J, Zhao B, Dong D, Li T, Cheng X, Gong W, Wang J, Zhang J, Xin G, Yu Y, Lei YL, Black JD, Li Z, Wen H | 2024 | Inhibition of O-GlcNAc transferase activates cGAS-STING pathway | https://doi.org/10.5061/dryad.4mw6m90ks | Dryad Digital Repository, 10.5061/dryad.4mw6m90ks |

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
