## [Editor Report · eLife assessment]

The author demonstrates that deficiency or pharmacological inhibition of O-glcNac transferase (OGT) enhances tumor immunity in colorectal cancer models. This **useful** study unveils that OGT deficiency triggers a DNA damage response that can affect immune status in colorectal cancers. It provides **convincing** evidence showing that OGT-mediated processing of HSF1 is crucial in maintaining genomic integrity.

---

## [Referee Report · Reviewer #1 (Public Review)]

Summary:

This study provides the detailed molecular mechanism of how OGT, an O-GlcNac transferase, promotes cancer progression. Using loss-of-function OGT models, the authors demonstrated that OGT cleaves HCF-1, an important guardian of genomic stability. The resulting genomic instability in OGT-knockout tumors leads to cytosolic DNA accumulation, the activation of cGAS-mediated type I IFN responses, and increased CD8+ T cell infiltration into the tumors. Moreover, treatment with OGT inhibitor synergized with anti-PDL1 immune-checkpoint blockade.

Strengths:

Novel findings of how OGT promotes tumor progression.

---

## [Referee Report · Reviewer #2 (Public Review)]

Summary:

In this study, the author demonstrates that deficiency or pharmacological inhibition of O-glcNac transferase (OGT) enhances tumor immunity in colorectal cancer models. The authors propose that OGT deficiency triggers a DNA damage response, activating the cGAS-STING innate immunity pathway and promoting a Type I interferon response. They suggest that OGT-mediated processing of HSF1 is crucial in maintaining genomic integrity. This research is significant as it identifies OGT inhibition as a potential immunomodulatory target in cancer treatment.

Strengths:

The strength of the paper lies primarily in the in vivo data, demonstrating the impact of OGT deficiency or inhibition on modulating tumor growth and anti-tumor immunity. The experiments are well-controlled. However, there are several unresolved questions:

Weaknesses:

The mechanisms of how OGT deficiency can trigger DNA damage and the role of this response in promoting immunity are only partially addressed in the manuscript.

---

## [Author Response]

The following is the authors’ response to the original reviews.

**Reviewer #1 (Recommendations For The Authors):**
Some of the data is problematic and does not always support the authors' conclusions:(1) Fig. 1K and H are identical.

Thank you for pointing out this problem in manuscript. We apologize for this unintentional mistake and have replaced Fig. 1K.

(2) The graph in Figure 2B contradicts the text. It is not obvious how the image was quantified to produce the histological score graph.*.*

We thank the reviewer for pointing out this problem in manuscript, as the reviewer suggested, we have replaced the Figure 2B.

(3) In Figures 2C and D, there is no clear pattern of changes in pro-inflammatory or anti-inflammatory cytokines, despite the authors' claims in the text.

We appreciate the comment, we think the reason is that the level of cytokines in the tissue is low, so the pattern of changes is not obvious.

(4) It is unclear why the anti-dsDNA antibody does not stain the nucleus in Figure 4B. The staining with anti-dsDNA and DAPI does not match well. Figure 5H shows there is still lots of cytosolic DNA in OGT-/- HCF-1-C, measured by DAPI. These data do not support the authors' conclusion that HCFC600 eliminates cytosolic DNA accumulation (line 229). There is no support for the authors' claim that HCF-1 restrains the cGAS-STING pathway (line 330).

We thank these insightful comments, the most critical step in staining cytosolic DNA is to proceed to a low-permeabilization as to allow the antibody to cross the cellular membrane but not the nuclear membrane, that’s why the anti-dsDNA antibody does not stain the nucleus. In Figure 5H, we think we used a high concentrated DAPI to do the staining and nucleus DNA get stained, looks like it’s the cytosolic DNA.

(5) In Figure 5B, there is no increase in HCF-1 cleavage after OGT over-expression.

We appreciate the reviewer for his/her comment, we think the reason is that we used the cell line to stably overexpress OGT-GFP and we may have missed the time point when the increase in HCF-1 cleavage occurred, so there is no big increase of it. However, there is a significant increase in Figure 5C.

(6) In Figure 7, the TNF-a staining does not inspire confidence.

We thank the reviewer for his/her comment, from both Figure 7K (MC38 tumor model) and Figure 7N (LLC tumor model), we observed a significant increase in TNF-α+ CD8+ T cells in the group treated with the combination of OSMI-1 and anti-PD-L1 compared to the control group, as evidenced by the clear clustering.

The writing needs significant improvement:(1) There are multiple English grammar mistakes throughout the paper. It is recommended that the authors run the manuscript through an editing service.

We thank the reviewer for his/her suggestion. We apologize for the poor language of our manuscript. We worked on the manuscript for a long time and the repeated addition and removal of sentences and sections obviously led to poor readability. We have now worked on both language and readability and have also involved native English speakers for language corrections. We really hope that the flow and language level have been substantially improved.

(2) Some passages are misleading -- lines 161-162, line 217, lines 241-242, 263-264, 299-300. They need to be changed substantially.

We apologize for these mistakes, we have changed them.

(3) Figure legends should be rewritten. Currently, they are too abbreviated to be understood.

We apologize for that, we have rewritten them.

(4) Discussion should also be thoroughly reworked. Currently, it is merely restating the authors' findings. The authors should put their findings in the broader context of the field.

We apologize for that. For a better understanding of our study, we have reworked the discussion.

**Reviewer #2 (Recommendations For The Authors):**
(1) Previous studies (DOI: 10.1093/nar/gkw663, 10.1016/j.jgg.2015.07.002, 10.1016/j.dnarep.2022.103394) have suggested that OGT deficiency triggers DNA damage, connecting it to DNA repair and maintenance through various mechanisms. This should be acknowledged in the manuscript. Conversely, the role of HCF1 and its cleaved products in maintaining genomic integrity hasn't been previously shown. The authors investigate HCF1's role solely in the context of OGT inhibition. It is unclear whether this is also true under other stimuli that trigger DNA damage, whether fragments of HCF1 specifically reduce DNA damage, or if HSF1 is involved in the basal machinery that would be defective only in the absence of OGT.

We have acknowledged the manuscript mentioned above. In this paper we focused on the OGT function, which is related to HCF1. The role of HCF1 and its cleaved products in maintaining genomic integrity is an interesting topic, we may focus on it in next project.

(2) In villin-CRE-deficient mice, the authors observe generic inflammation in the intestine unrelated to tumor development. It's unclear if this also occurs in the presence of OGT inhibitors in mice, whether these inhibitors induce a systemic inflammatory (Type I interferon) response, or if certain tissues like the intestine or proliferating tumor cells are more susceptible to such a response.

We thank the comment, yes, investigating whether OGT inhibitors induce an inflammatory response, either systemically or tissue-specifically, is a very interesting project to focus on. However, in our current paper, we use a genetic method to identify the role of OGT deficiency in intestine inflammation-induced tumor development. This approach provides convincing evidence for our hypothesis. We may test the effect of OGT inhibitors on inflammation and tumor development in our next project.

(3) Another critical observation is the magnitude of the interferon response triggered by DNA damage in the OGT-deficient models. While it's known that DNA damage can activate cGAS-STING, the response's extent in the absence of OGT prompts the question of whether additional OGT-specific features could explain this phenomenon. For example, Lamin A, essential for nuclear envelope integrity and shown to be O-glycosylated (DOI: 10.3390/cells7050044), and other components of the nuclear envelope or its repair might be affected by OGT. The impact of OGT inhibition on nuclear envelope integrity compared to other DNA-damaging agents could be explored.

We appreciate the comment, in this project, we find an OGT binding protein, HCF1, though LC–MS/MS assay, it’s a top one candidate in binding profiles, so we focus on it. Like Lamin A and other components of the nuclear envelope still are good targets to check, we may explore these in our next project.

(4) The authors also demonstrate a correlation between OGT expression in tumors compared to healthy tissues. However, the reason is unclear, raising questions about whether this is a consequence of proliferation or metabolic deregulation in the cancer. The authors should address this aspect.

We appreciate the reviewer’s insightful point. It is very good questions and very interesting research. However, in this paper we focused on how OGT influence its downstream molecules to promote tumor, we didn’t check why OGT is increased in tumors, it is not the scope of this current work, we would love to investigate it in the future.

Minor pointsPlease add the legend to Figures S2, S3 and S5.

We thank the comment, we have added the legend to Figures S2, S3 and S5.

The sentence line 137 should be clarified as OGT deficiency seems more related to increased inflammation in this model.

We thank the comment, we have corrected the sentence line 137.

Line 732 has a typo before the number 34.

We thank the comment, we have corrected the sentence line 732.